# Integrated Management of the Cotton Charcoal Rot Disease Using Biological Agents and Chemical Pesticides

**DOI:** 10.3390/jof10040250

**Published:** 2024-03-26

**Authors:** Ofir Degani, Assaf Chen, Elhanan Dimant, Asaf Gordani, Tamir Malul, Onn Rabinovitz

**Affiliations:** 1Plant Sciences Department, MIGAL—Galilee Research Institute, Tarshish 2, Kiryat Shmona 1101600, Israel; elhanand@migal.org.il (E.D.); asigordani1@gmail.com (A.G.); tamir1012011@gmail.com (T.M.); onnrab@gmail.com (O.R.); 2Faculty of Sciences, Tel-Hai College, Upper Galilee, Tel Hai 1220800, Israel; assafc@migal.org.il; 3Soil, Water and Environment Department, MIGAL—Galilee Research Institute, Tarshish 2, Kiryat Shmona 1101600, Israel

**Keywords:** Azoxystrobin, crop protection, disease control, field study, integrated control, *Macrophomina phaseolina*, *Trichoderma*, Real-Time PCR, remote sensing

## Abstract

Charcoal rot disease (CRD), caused by the phytopathogenic fungus *Macrophomina phaseolina*, is a significant threat to cotton production in Israel and worldwide. The pathogen secretes toxins and degrading enzymes that disrupt the water and nutrient uptake, leading to death at the late stages of growth. While many control strategies were tested over the years to reduce CRD impact, reaching that goal remains a significant challenge. The current study aimed to establish, improve, and deepen our understanding of a new approach combining biological agents and chemical pesticides. Such intervention relies on reducing fungicides while providing stability and a head start to eco-friendly bio-protective *Trichoderma* species. The research design included sprouts in a growth room and commercial field plants receiving the same treatments. Under a controlled environment, comparing the bio-based coating treatments with their corresponding chemical coating partners resulted in similar outcomes in most measures. At 52 days, these practices gained up to 38% and 45% higher root and shoot weight and up to 78% decreased pathogen root infection (tracked by Real-Time PCR), compared to non-infected control plants. Yet, in the shoot weight assessment (day 29 post-sowing), the treatment with only biological seed coating outperformed (*p* < 0.05) all other biological-based treatments and all Azoxystrobin-based irrigation treatments. In contrast, adverse effects are observed in the chemical seed coating group, particularly in above ground plant parts, which are attributable to the addition of Azoxystrobin irrigation. In the field, the biological treatments had the same impact as the chemical intervention, increasing the cotton plants’ yield (up to 17%), improving the health (up to 27%) and reducing *M. phaseolina* DNA in the roots (up to 37%). When considering all treatments within each approach, a significant benefit to plant health was observed with the bio-chemo integrated management compared to using only chemical interventions. Specific integrated treatments have shown potential in reducing CRD symptoms, such as applying bio-coating and sprinkling Azoxystrobin during sowing. Aerial remote sensing based on high-resolution visible-channel (RGB), green–red vegetation index (GRVI), and thermal imaging supported the above findings and proved its value for studying CRD control management. This research validates the combined biological and chemical intervention potential to shield cotton crops from CRD.

## 1. Introduction

Cotton (*Gossypium* spp. L.), serving as a source of income for ca. 100 million farmers, is the world’s primary source of natural fibers, boasting a cultivated area exceeding 33 million hectares in 2019 [1,2]. The cotton industry spans around 150 countries and stands out as a crucial annual crop with significant economic gains. It was noted that eight out of the top ten cotton-producing countries belonged to the category of developing nations (in 2017, according to Jans et al. (2021) [2]). While cotton is already well suited for growth in temperate, subtropical, and tropical climates globally, its future expansion may face obstacles due to the impacts of climate change [3]. Such global changes can enhance the distribution and negative effects of soil-borne diseases such as cotton charcoal rot (CRD) [4]. 

Biotic stress from various pathogens and pests significantly harms cotton crops, increasing global production costs for implementing control measures. Key diseases adversely affecting cotton include bacterial and fungal infections targeting leaves, stems, roots, and fruits [1]. *Macrophomina phaseolina* (Tassi) Goid, the CRD causal agent, is responsible for causing diseases in more than 500 cultivated and wild plant species across approximately 75 families [5]. The severity of the disease is linked to the degree of soil infection, variations in the fungus’s pathogenicity, and the specificity of the host [6]. In cotton, symptoms appear in the later stages of growth (during flowering and fruit development), including drying leaves and stems, wilting, and, ultimately, plant death [7]. *M. phaseolina* may cause a total yield loss in patches in the infected field [8].

The pathogen typically enters the roots by exploiting cracks in the phelloderm, resulting from dense soils’ expansion and cracking in hot, dry weather [4]. Preceding the irrigation timing can result in better growth, yield, and health than late spring watering [9]. The fungal uses cell-wall-degrading enzymes to penetrate and kill the host plant cells [10]. One common CRD symptom involves vascular bundles browning resulting from toxins interacting with a plant-response product, which is likely gossypol [4]. Gossypol, a significant sesquiterpene phytoalexin in cotton, is essential for defending the plant against invading pathogens. Early signs of the disease, usually in early summer in young developed plants, involve leaf yellowing, which is likely caused by toxins the pathogen produces during root colonization [11]. These toxins are then transported to the above-ground parts of the plants. Subsequently, despite irrigation, the plant may show symptoms of dehydration [4]. Recent reports indicate that the incidence of the disease has risen in cotton fields in Israel, which is potentially attributed to the shift to a more susceptible extra-long staple (ELS) Pima-type (*Gossypium barbadense*) variety and global warming [4].

The persistent survival of *M. phaseolina* in the soil and crop residues is primarily attributed to multicellular sclerotia, which are formed in the host tissues and are released into the ground as tissues decay [12,13,14]. Those asexual reproduction bodies make CRD management a significant scientific challenge. Various methods exist for managing the associated disease, including using resistant varieties, crop rotation, cultural practices, soil solarization, and soil moisture balancing [13]. However, these approaches demand much investment and extended periods for effectiveness, and other challenges impair their efficacy. The crop cycle is not deemed efficient due to the fungus’s strong saprophytic ability [12,15]. Since there is a lack of plant varieties resistant to highly aggressive pathogen strains, systemic fungicides have become the primary method of reducing their presence [16]. 

Several fungicides, including Carbendazim, Difenoconazole, Benomyl, and Azoxystrobin, have undergone testing against the pathogen, demonstrating varying degrees of control potential. A study by Bashir et al. [16] identified Carbendazim and Thiophanate methyl as effective treatments. In Israel, Cohen et al. [4] evaluated the efficacy of chemical control in enhancing plant resistance to *M. phaseolina*. When Azoxystrobin and Signum (Boscalid, 26.7% + Pyraclostrobin, 6.7%) were applied as a seed coating, they effectively prevented pathogen penetration for only 12 days. After this period, roots extended beyond the protected area, making them susceptible to pathogen penetration. A separate experiment assessed disease progression at 1, 2, and 3 months after seeding following three fungicide applications [4]. The Azoxystrobin treatment resulted in a 15% disease incidence, while the non-treated control and treatments with Prochloraz and Prothioconazole exhibited a disease incidence of approximately 30%.

However, as the public worries about increasing synthetic chemicals use and fungicide-resistant fungal strains becoming a serious problem [17], it is crucial to investigate alternative, eco-friendly methods for controlling the CRD pathogen. Managing soil- and seed-borne diseases through biocontrol is an effective strategy [18]. Using agriculturally significant microorganisms for disease management offers a safer alternative than traditional practices that harm the environment and agroecosystem and pose potential health risks. Indeed, plants that coexist with *Trichoderma* species exhibit improved growth and the suppression of various diseases compared to plants grown without these fungal biocontrol agents [19,20,21,22,23]. So, these antagonistic organisms offer multiple disease protection. 

The communication between the host plant and *Trichoderma* spp. leads to *Trichoderma*’s colonization of the rhizosphere region both externally and internally. While some *Trichoderma* species perform as opportunistic symbionts, others are considered to use a non-root mode of entry into the plant and dwell as true endophytes [24], protecting plants from abiotic stresses and diseases by inducing transcriptomic changes in plant [25,26]. Additionally, these organisms have evolved different strategies to combat pathogens, such as producing antibiotics, competing for nutrients, and engaging in mycoparasitism. Nevertheless, the practical use of this technology in agriculture is frequently impeded by natural challenges, leading to unpredictable control outcomes [27,28,29].

Applying biological control agents, with or without organic amendments or following solarization, is considered a practical method for managing *M. phaseolina* [13]. In a prior study, we examined the effectiveness of eight *Trichoderma* isolates against the *M. phaseolina* pathogen. Among these, *T. asperellum* (P1) and *T. longibrachiatum* (T7407, T7507) were identified as promising treatments for CRD control in seedlings grown in pots [30]. The T7407 treatment led to significant enhancements in plant survival (34%), wet weight (45%), plant height (32%), and phenological development (56%) after 42 days compared to the control. Simultaneously, it reduced pathogen root infection to near-zero levels [30]. These findings from the seedling stage suggest the potential for further research to implement this approach in field conditions throughout an entire growing season, particularly given the tendency of the disease to manifest during plant maturation.

Experiments that tested chemo-agents and bioagents separately were performed at various levels, including field assay, over the years. For example, eight fungi-toxicants and *Trichoderma* species were tested independently against the CRD pathogen under in-vitro conditions [8]. Among them, Carbendazim and MEMC (Emisan-6) at 10 ppm concentration and *T. viride* were the most effective in reducing the pathogen growth. A recent scientific effort suggests a control approach aims to uphold the efficacy of chemicals while substantially decreasing their amounts [31].

This research line focuses on combining the benefit of microorganisms with low dosages (or low toxicity) of chemical fungicides to protect cotton plants from CRD over the growing season [9]. Integrating biological agents with chemical pesticides presents several potential benefits [32]. These advantages encompass diminishing the development of resistance, promoting environmental friendliness, sustaining the stability of bio-shielding treatments in fluctuating environmental conditions, and achieving heightened efficiency, which is particularly crucial in severe disease cases [33].

In a previous study, we applied an integrated pest management approach, combining a *Trichoderma*-based biological treatment with Azoxystrobin irrigation to minimize cotton CRD in a semi-field (open-enclosure) full-season potted experiment [9]. The eco-friendly treatments with *T. asperellum* (P1) and *T. longibrachiatum* (T7507) alone protected cotton plants’ growth and enhanced health indexes throughout the season. Adding minimal Azoxystrobin concentration improved T7507 impact but impaired P1 efficiency. By the harvest (day 173), Real-Time PCR (qPCR) monitoring of pathogen DNA in plant roots showed that the combined treatments of *T. longibrachiatum* (T7407, T7507) + Azoxystrobin reached a high efficacy level of 86–91% pathogen repression. 

In the commercial field, the biological and chemical pesticides were tested separately [9]. The highest yields came from treating seeds with a *Trichoderma* species blend (P1, T7407, *Trichoderma* sp. O.Y. 7107 isolate). This bio-shielding outperformed Azoxystrobin treatment in yield improvement and was similar to this conventional treatment in reducing *M. phaseolina* infection. This initial essential field trial opened the door to the future testing and application of integrated biological–chemical management on such a scale.

The current study intended to establish an integrated control method to protect cotton plants against charcoal rot pathogen using selected *Trichoderma* fungal strains, which demonstrated success in lab and seedlings tests [30] and entire season trials [9], and a previously successfully tested systemic chemical fungicide, Azoxystrobin [9]. Our prior research confirmed the tolerance of the *Trichoderma* species to Azoxystrobin, even at relatively high compound concentrations [33]. The study tests the combined capacity of bio-protective species and fungicides to control cotton CRD under both growth room and field conditions. 

To this end, a mixture of three *Trichoderma* species was used to coat cotton seeds, and the treated seeds were tested as a sole treatment or combined with Azoxystrobin sprinkling at sowing or by irrigation during later growth stages, in pots, under controlled conditions, and in a commercial field. The plants’ growth, yield, and health tracking were accompanied by a molecular method targeting the DNA of the CRD agent, *M. phaseolina*, in the plant roots (using qPCR detection). At the season’s end, the field experiment was additionally monitored using remote sensing techniques, including high-resolution visible-channel (RGB, red, green, blue) and thermal aerial imaging. 

## 2. Materials and Methods

### 2.1. Fungal Sources and Growth Conditions

In 2017, the phytopathogenic fungus *M. phaseolina* (isolate Mp-1) was isolated from cotton plants that exhibited fungal infection at Roni Cohen’s lab in the Newe Ya’ar Research Center, Newe Ya’ar, Israel. This isolate was identified using multiple methods, including assessments of pathogenicity, physiology, colony morphology, microscopic characteristics, and molecular traits [7,30]. The *Trichoderma* species chosen for this study had previously demonstrated effectiveness against the CRD agent, *M. phaseolina* [9,30]. These species are *Trichoderma* sp. O.Y. T7107, *T. Longibrachiatum* (T7407), and *T. asperellum* (P1) (Table 1).

All fungal species were kept in darkness on a potato dextrose agar (PDA) medium at 28 ± 1 °C with high humidity for 4–7 days. Following the manufacturer’s instructions, the PDA medium was made by dissolving 39 g of PDA powder in 1 L of double-distilled water (DDW). To subculture the fungus, a 6 mm diameter agar disc from the edge of the culture was transferred to a new Petri dish containing fresh PDA, following the procedure described in [30]. The Petri plates were placed in the MaxQ™ 6000 Incubated/Refrigerated Stackable Shakers (Thermo Fisher Scientific, Waltham, MA, USA) and incubated at 28 °C in the dark. In submerged cultures, five fungal discs were cultivated in 150 mL potato dextrose broth (PDB) within a 250 mL Erlenmeyer flask. To prepare the PDB medium, 24 g of PDB powder was dissolved in 1 L of distilled water. The flasks were sealed with a breathable stopper and shaken at 150 rpm for 7–10 days under the same conditions.

### 2.2. Pot Assay in a Growth Room

#### 2.2.1. The Trial Architecture

The study was conducted in pots in a growth room with the Goliath V-6 cotton cultivar (extra-long staple (ELS) Pima-type, *G. barbadense*, from Israel Seeds). The experiment included two types of seed coating (biological or chemical) as controls and five different applications of the fungicide Azoxystrobin (Amistar S.C.; Syngenta, Basel, Switzerland, supplied by Adama Makhteshim, Ashdod, Israel), in combination with each type of seed coating, for a total of 12 treatments. Such treatments are listed in Table 2 and shown in Appendix A, along with the dosage of fungicide and time of application. The chemical seed coating was performed by Israel Seeds Company with a standard fungicide mix (thiram, captan, and metalaxyl-M), which does not protect from CRD. The biological seed coating was performed with a mix of spores, mycelium fragments, and extrolites from the different Trichoderma cultures, as described in Section 2.2.2. The treatments were compared to two additional mock groups of non-infected plants that underwent solely biological or chemical seed coating. All experiment groups were conducted in seven biological replications (84 pots in total). The specific repeat number used to analyze the results is indicated in each figure. The treatments and control pots were arranged in the growth room in a fully randomized design.

#### 2.2.2. The Biological Protective Treatment

The *Trichoderma* isolates were cultivated in a liquid PDB medium, as explained in Section 2.1. The cultures were ground in a blender for 2 min to obtain a mixture of spores and small mycelial fragments (colony-forming units, CFUs). This mixture of growing media and mycelia was utilized to make the solution for seed coating, as described before [9]. A 9-liter tank was prepared for the *Trichoderma* species mixture (3 L of each species, T7107, T7407, P1, Table 1). Spores and mycelium fragments were also harvested from 15 solid PDA fungal colonies (five of each of the three species in the mix). They were transferred to the container to enrich the suspension of CFU and extrolites. Tween 80 (in a final concentration of 0.05%) was added to the container to facilitate the fungus mycelium’s adhesion to the seeds. After thoroughly mixing, 15 kg of cotton seeds (not chemically treated) was added to the liquid. After mixing the seeds by hand with the *Trichoderma* mix solution for ten min, the seeds were removed from the liquid, spread out on a plastic sheet, and allowed to dry for 24 h. The bio seed-coating solution contained 3.27 × 10^6^ CFU/mL. After conducting the above coating protocol, the coating solution runoff had 1.19 × 10^6^ CFU/mL, so 2.08 × 10^6^ CFU/mL adhered to the seeds. The liquid-to-cotton seeds ratio is 0.6 mL suspension to 1 g of seeds. Each dry cotton seed weighs ca. 125 mg, so 1 g contains approximately eight plant seeds. Therefore, each seed was treated with a ca. 75 µL coating solution. The pH of the coating solution was dropped from 4.43 before the procedure to 1.68 in the residual liquid. Still, inspecting the germination of 60 seeds (after two days of incubation at 28 ± 1 °C) showed that the coating procedure caused only a slight reduction in germination from 88% to 82%.

#### 2.2.3. The Growth Conditions

Pots (2.5 L) were filled with heavy soil from Israel’s Northern R&D experimental farm (Hula Valley, Upper Galilee, 33°09′08.2″ N 35°37′21.6″ E) to mimic as much as possible field conditions. This soil has no record of CRD, and if such infection occurred, it was assumed to be minimal. The soil was mixed with 33% Perlite (No. 4) for aeration. At a depth of 5 cm below the soil surface, we spread 10 grams of Osmocote 14-4-28, a 4-month slow-release fertilizer (Israel Chemicals Ltd., ICL, Tel Aviv, Israel), and 30 grams of sterilized *M. phaseolina* infected millet grains (as will describe below). In each pot, five cotton seeds were sowed to a depth of ca. 3 cm. The plants were grown in a growth room in an artificial light regime of 16 h and 8 h of night with 58–82% humidity at 25 ± 2 °C. Each pot was watered every 24 h for 5 min at a dripping rate of 35 mL/min of tap water using a computerized drip lines system. After the thinning to one plant per pot (day 29), the irrigation was reduced to 5 days a week (excluding Fridays and Saturdays). The experiment concluded after 52 days.

#### 2.2.4. *Macrophomina phaseolina* Infection

Soil inoculation was performed by adding 10 g of sterilized millet seeds per pot to the upper layer of the soil (5 cm below the surface). These seeds were previously infected with *M. phaseolina*, according to a preparation method detailed in reference [33]. The inoculation process took place one week before sowing. Following the above soil surface emergence, 1 and 2 weeks after sowing, additional soil infections were carried out on all treatment pots. These 2nd and 3rd complementary infections were conducted by adding three culture discs of *M. phaseolina* (each with a 6 mm diameter) near each sprout (ca. 2.5 cm depth).

#### 2.2.5. Experimental Determinations

The above-ground emergence percentages in each treatment were checked after ten days. Survival percentages (counting the living sprouts), growth indices (roots and shoot weight, plants above ground height, and the leaves number), and fungus DNA content in each plant’s root using qPCR were determined on days 29 and 52 past sowing (detailed in Section 2.4). In the molecular analysis, very low threshold reads were received in the control farm soil’s uninfected plants. This is probably attributed to the minor presence of *M. phaseolina* in that soil. Thus, the average read of the healthy control groups was subtracted from all the experiment reads. Table 3 provides specific dates of significance of the growth room bio-chemo control experiment.

### 2.3. Commercial Field Trial

#### 2.3.1. The Trial Architecture

The biological-based integrated management was studied compared to the typical chemical approach in a commercial cotton field. The same bio-coated seeds batch used for the growth room trial was used for the field assay (see Section 2.2.2 for the seeds’ coating detailed preparation protocol). The standard chemical intervention relies on general pest control using Captan seed coating and, in severe CRD-infected fields, adding Azoxystrobin or other fungicides to the sowing strip or during the season through irrigation. The experiment was performed in spring–summer 2023 in the kibbutz Hulda commercial field in the Shephelah region, Israel, at 31°49′23.1″ N 34°52′32.2″ E, which is part of a sizeable profitable field. In recent years, the area has been affected by charcoal rot disease [9]. The experiment was performed in a randomized complete block design using the Goliath V-6 cotton cultivar. As in the growth room (Table 2), the 12 treatments include chemical seed coating and biological seed coating (a mix of *Trichoderma* species (T7107, T7407, and P1, Table 1). The chemical sprinkling in the sowing strip with the seeding used Azoxystrobin (Mirador 250 SC, 250 g/L active ingredient, Adama Makhteshim, Airport City, Israel) at 100 mL per 0.1 ha. Additionally, two chemical treatments with this fungicide at 200 and 400 mL per 0.1 ha were employed through a drip irrigation line 44 and 78 days from sowing.

Each treatment group included eight replications (plots). However, to prevent outliers (irregular growth) due to various cultivation issues, especially in the field margins (such as weeds, wind, and inconsistent water dissemination), 3–6 plots per treatment were sampled during harvest. The precise number of samples analyzed is indicated in the legend of the individual figures. Each experimental plot comprised six rows (in a garden bed, a raised area of soil above the surrounding ground) and measured 6 m wide and 27 m long with 0.96 m of row spacing. The field’s length from south to north is about 200 m, and its width from east to west (12 garden beds) is about 72 m. The total area included 96 plots and was about 1.44 ha. 

#### 2.3.2. Growth Protocol and Conditions

On 9 April 2023, the seeding process was carried out by sprinkling 100 L of water (or Azoxystrobin solution in some treatments) per 0.1 ha onto each sowing strip to ensure even germination. The sowing depth was 5 cm with approximately 13.3 plants per meter. After two weeks, an evaluation of above-ground emergence was conducted, showing normal sprouting across most areas of the field. Precise dates of importance are present in Table 4.

To maintain homogeneousness in soil moisture, improve fungicide movement in the ground, and reduce soil cracking and damage to the roots [35], assumably leading to plant phelloderm injury and facilitating *M. phaseolina* root colonization [4], we placed drip lines on the bare soil tread paths between the rows. Irrigation was initiated on day 43, applying a total water dose of 40 mm/day. The irrigations were performed twice a week. The total amount of irrigation was ca. 570 mm. Penman evaporation estimation was ca. 1000 mm over the entire growing period (9 April–7 September). Fertilizers and treatments against various pests were applied throughout the experiment to mitigate risks other than *M. phaseolina* infection. The temperature and humidity conditions measured during the growth period were normal and suitable for CRD development (Appendix A).

#### 2.3.3. Symptoms and Yield Estimation

At 74 and 151 days post-sowing, root samples were collected (one typical healthy plant from each replication, a total of 7–8 plants per treatment) for DNA extraction. The disease impact was studied before the irrigation ended, on day 151, by counting the number of dead plants in the two middle rows of each experimental plot. The yield was assessed on day 210 by harvesting the two center rows in the above repeats using a cotton-picking machine.

### 2.4. Molecular Real-Time PCR Diagnostic

#### 2.4.1. DNA Extraction

The roots of the plants were cut into portions about 2 cm in size after being cleaned once with tap water and flowing sterile DDW. Every repetition’s total weight was adjusted to 0.7 g. To extract the DNA, the root sample and 4 milliliters of cetyltrimethylammonium bromide (CTAB) buffer solution were placed into a BioMed bag (universal extraction bags, Bioreba, Reinach, Switzerland). An electric tissue homogenizer (Bioreba, Reinach, Switzerland) was used to homogenize the material for five min. As previously illustrated [36] with slight adjustments [30], the homogenized samples were subjected to DNA purification. The DNA was suspended in 100 µL of HPLC-grade pure water and stored at −20 °C until qPCR testing was used.

#### 2.4.2. qPCR Technique

Here, we employed the previously published protocol for the qPCR process [7,30] using an Applied Biosystems apparatus with 384-well plates (Foster City, CA, USA) and the ABI-7900HT model. The technique is based on a traditional qPCR process created to identify mRNA (cDNA), but it has been optimized to detect *M. phaseolina* DNA effectively. A total of 5 µL of the reaction mixture (2.5 µL of iTaq^TM^ Universal SYBR^®^ Green Supermix; Bio-Rad Laboratories Ltd. (Rishon Le Zion, Israel), 2 µL of sample DNA extract, and 0.25 µL of each of the primers, forward and reverse, 10 µM each) was added to each sample well.

The qPCR procedure included a pre-cycle activation phase (1 min at 95 °C), 40 cycles of denaturation (15 s at 95 °C), annealing and extension (30 s at 60 °C), and a melting curve analysis. Four tests and analyses of a sample of plant roots were performed to guarantee the consistency of those technical repeats. The MpK (FI/RI) primers, which are sequences listed in Table 5, were specifically used to detect *M. phaseolina*. The quantity of DNA was normalized using the COX reference gene (Cox primers are in Table 5), which is found in both the pathogen and the plants [37]. This “housekeeping” gene encodes for the final enzyme in eukaryotic mitochondria’s respiratory electron transport chain [37]. Following the reaction, the ΔCt model was employed to ascertain the relative quantity of the target *M. phaseolina* fungal DNA [38,39]. It was assumed that all samples had the same level of efficiency.

### 2.5. Aerial Imaging

Thermal and visible light (RGB) sensing were used to monitor and estimate the CRD’s progression at the commercial field experiment using aerial imaging instruments onboard unoccupied aerial vehicles (UAVs). The imaging campaign occurred on 18 September 2023 at day 162 from sowing. RGB imaging was conducted using the built-in L1 camera onboard the Matrice 300 RTK UAV (DJI, Shenzhen, China) at 30 m above ground level (AGL) with a 0.67 cm pixel size. Thermal imaging was conducted using a Flir A655SC camera (Flir, Wilsonville, OR, USA), 640 × 480 pixels, onboard a DJI Matrice 600 pro-UAV (DJI, Shenzhen, China) at 50 m AGL with a 3.46 cm pixel size. Thermal and RGB imaging campaigns were conducted using the Pix4D Capture (Pix4D, Prilly, Switzerland) and UgCS (Version 4.21, SPH engineering, Riga, Latvia) pre-programmed flightpath control software, respectively; an 80% overlap (forward and side) was chosen to create ortho-mosaics. Pix4Dmapper software (Version 4.8.4, Pix4D, Prilly, Switzerland) was used for mosaicking. The green–red vegetation index (GRVI) was applied to the RGB ortho-mosaic to detect plant vigor and stress and locate plants affected by the CRD [42].

### 2.6. Statistical Analysis

Analysis and statistical processing were performed using GraphPad Prism software, version 10.1.0 (316) 17 October 2023 (GraphPad Software Inc., San Diego, CA, USA) and JMP pro software, version 16 (SAS Institute Inc., Cary, NC, USA). The data were analyzed using a two-way variance analysis (ANOVA) at a significance level of *p* < 0.05. High variability in the results of the growth room pot assay (high standard error values) is expected, considering the objective difficulty of uniformly inoculating the plants. In field trials under natural soil infection conditions, such a scenario is probable, and the problem worsens. Consequently, it is challenging to obtain statistically significant differences. To maintain the power of the test and reduce the risk of type II errors, we opted not to adjust for multiple tests and used Fisher’s least significant difference (LSD).

## 3. Results

### 3.1. Pot Assay in a Growth Room

The present study applied an integrated biological–chemical control method based on selected *Trichoderma* species and Azoxystrobin fungicide to control the CRD causal agent, the fungus *M. phaseolina*. The efficacy of the integrated approach was first evaluated under a controlled environment in young plants (up to the age of 52) in a growth room (Figure 1).

Ten days from sowing, the emergence percentage was between 89% and 100%. A deeper examination of the data uncovers a slower emergence of the biologically coated seeds (93.5%) than chemically treated (98.8%) without a statistically significant difference. Because the disease often remains latent in sprouts, it is difficult to achieve statistically significant differences in the growth parameters of the plants, as was the case here. Looking at the growth parameters measured on day 29 post-sowing (Figure 1A–C), the integrated approach never performed worse than the chemical approach, according to statistical analysis (Figure 2A–F, Appendix A). Among the seeds chemo-coating treatments, one treatment that stood out was the Azoxystrobin sprinkling at seeding (Sp), which improved the sprouts’ root weight by 21% compared to the non-infected control (Figure 2C). In contrast, negative effects in the chemical seed coating group are highlighted on shoot weight and height and the number of leaves due to Azoxystrobin irrigation (Sp + D200 and Sp + D400 treatments, *p* < 0.05 compared to seed coating only, Figure 2A,B,E).

Interestingly, for the number of leaves, the treatment (Sp + D200) was significantly less harmful when combined with the biological seed coating than with the chemical coating (*p* < 0.05, Figure 2E). Remarkably, in the shoot weight assessment, the bio-seed-coating-only treatment outperformed (*p* < 0.05) all biological-based treatments and all Azoxystrobin-based irrigation treatments. This biological coting also achieved statistically discernible improvement in the leaf count values compared to the biological mock control, the D400 and the Sp + D200, and sp + D400 treatments (Figure 2E).

At the end of the experiment, on day 52 (Figure 1E,F), the growth indices were generally similar to those recorded on day 29 but had less statistical significance (Figure 3, Appendix A). As for day 29, the treatments Sp + D200, combining chemical coating with Azoxystrobin sprinkling and irrigation, exerted a negative effect on plant development, significantly reducing shoot weight and height, and leaves number compared to chemical SC only (Figure 3A,B,E).

The *M. phaseolina* DNA variations in the plants’ roots (sampled at midseason, day 29, and at the end of the experiment, day 52) after the different treatments are reported in Figure 4 and Appendix A. Regarding day 52 data, there were 4–5 repetitions in a few treatments due to outlier substruction (identified using the ROUT method in GraphPad software with the Q = 1% recommended setting) [43]. This restriction leads to a lesser statistical analysis power. On day 29, for the root infection, as already observed for the plant growth parameters, the biological and integrated approach performed similarly to the chemical approach. The treatment Sp + D400 increased the pathogen infection in combination with a chemical coating on day 29 and in combination with a biological coating on day 52. On day 52, the chemical SC + D400 was more protective than the corresponding bio-coating + D400. The same occurred for chemo-coating + Sp + D400, which was superior to the corresponding biocoating + Sp + D400.

### 3.2. Commercial Field Trial

The same treatments evaluated in young plants under the growth room-controlled environment were applied at a commercial field scale and assessed during an entire growth season (151 days, Figure 5). The plants developed well at the first growth stages (up to 74 days post-sowing), and the CRD symptoms were mostly latent (Figure 5A–C). In contrast, by maturation, severe disease outbreaks were documented in the less beneficial management practices (Figure 5D,E). Indeed, those plants had up to 83% dehydration compared to the most successful methods (as will be elaborated below). Patches of wilting plants in the field characterized the disease spreading. The typical CRD symptoms involving vascular bundle browning, resulting from toxins interacting with a plant-response product, likely gossypol, were observed (Figure 5D–F).

The yield assessment did not allow the identification of statistical differences due to the low number of repeats (3–6 plots per treatment; some field margin plots had irregular growth due to uneven water dispersal, weeds, wind, and other reasons). Biological treatments had an equal impact as the chemical intervention, increasing the cotton plants’ yield by up to 17% (Figure 6A, Appendix A).

Likewise, considering plant symptoms and root infection, the integrated approach in the field is generally not performed differently than the chemical approach. Some integrated treatments appeared promising in reducing CRD symptoms, like biochemical coating combined with Azoxystrobin sprinkling at sowing (Figure 6B, Appendix A, 27% more healthy plants). This biological coating + Sp significantly lowered symptoms compared to chemical coating + D200 and chemical coating + Sp + D200 (83%, *p* < 0.05). Still, this treatment (Bio + Sp) was not statistically different from all the other treatments, biological or chemical. In the near-harvest sampling, bio-coating + Sp + D400 reduced the pathogen DNA levels by 37%, with a statistically significant difference compared to chemical coating + Sp + D400, but was statistically equal to all the other treatments (Figure 7B). Analysis that refers to the field trial symptoms assessment presented in Figure 6B and considers all treatments together in each group revealed a significant advantage (*p* = 0.0433, paired *t*-test) to the bio-chemo integrated management over the traditional chemical intervention (chemical seed coating and applying Azoxystrobin by sprinkling with seeding and/or irrigation during the season), regarding the plants’ health. Also, a significant negative correlation (*p* = 0.0016) was found between the number of wilting plants (Figure 6B) and the yield (Figure 6A). Finally, a strong positive correlation (*p* = 0.0002) was identified at the season-ending between *M. phaseolina* DNA in the plants’ roots and the number of wilting plants.

The commercial field trial was imaged from the air on day 162 using UAVs. Disease signs and changes in plant foliage temperature can be seen in the high-resolution RGB, GRVI, and thermal aerial imaging (Figure 8, Table 6). Due to the high variability in these measures, no statistically significant difference between the treatments could be identified. Still, a significant negative correlation (*p* = 2.87 × 10^−5^) was found between the plots’ average temperature and vegetation vigor tracked by the GRVI index. Two representative plots, plot 44 on the left (chemical coating only) and plot 53 on the right (bio seed coating + Azoxystrobin sprinkling and irrigation at low dosage, Sp + D200), were selected in the middle of the field to demonstrate the CRD impact on those measures. Plot 53 (on the right) exhibited more vivid and developed vegetation (darker green, Figure 8A), a higher GRVI index (mean of 0.085, Figure 8B), and a lower foliar temperature evidenced by the darker purple color (mean of 35.5 °C, Figure 8C) compared to Plot 44 on the left (mean GRVI index of 0.075 and temperature of 36.6 °C).

## 4. Discussion

The *Macrophomina phaseolina* charcoal rot disease (CRD) is a significant worry to cotton growers worldwide. Consequently, it is a global scientific priority, especially given that the necrotrophic fungus has a wide host range and poses a major danger to numerous horticultural and agricultural plants [5]. Our toolkit for managing the disease is currently limited despite ongoing development and testing of CRD control techniques [44,45,46,47,48].

The current work focused on expanding this toolkit using a *Trichoderma*-based biocontrol strategy either alone or in combination with low-dosage Azoxystrobin. As an example for this research line, *Trichoderma harzianum* (strain SH2303) and a mixture of Difenoconazole–propiconazole were effectively used in tandem to manage the maize southern corn leaf blight agent, *Cochliobolus heterostrophus* [49]. Additionally, an integrated (bio-chemical) pest control method targeting *Magnaporthiopsis maydis* in corn [33] has demonstrated the promising potential of this approach. Combining a *Trichoderma* species mixture and low dosage Azoxystrobin reduces disease prevalence and enhances corn crop quantity and quality.

In cotton, certain rhizospheric bacteria, particularly those of the *Bacillus* species, are renowned among biocontrol agents. This reputation is attributed to their ability to produce antimicrobial metabolites, induce systemic resistance, and efficiently establish colonization in soil [50]. For example, two bacteria strains, *Bacillus megaterium* ZMR-4 and *B. subtilis* IAGS-174, along with Benzothiadiazole are used for the inhibition of CRD and growth promotion of cotton under greenhouse and field conditions [51]. In greenhouse experiments, combining the bacteria and chemical pesticides achieved 80–83% biocontrol over the pathogen, whereas individual benzothiadiazole treatments reached only 69–71%. When used separately, the bacteria strains suppressed the disease by 50–53%. In field conditions, plants treated with *B. megaterium* ZMR-4 and Benzothiadiazole exhibited the highest biocontrol efficiency, ranging from 72.8% to 78.9% [51].

The current study was pioneering in the examination of the CRD-integrated bio-chemo protocol on a commercial field scale in Israel. It is an essential step toward commercializing this management approach. The advantage of the bio-based integrated methodology is that it provides the *Trichoderma* agents with stability and strength in unstable environments [30,32]. A review by Ons et al. (2020) evaluated the integrated approach against distinct phytoparasitic fungi in various plants [31]. This strategy is advantageous since it lessens the impact of chemical treatments on health and the environment. It may also be useful in combating multiple soil fungal phytopathogens and is necessary to prevent the development of fungal resistance to fungicides [52,53,54].

The current and previous reports [9] present a two-year study on the same location (kibbutz Hulda commercial field in south Israel) that uncovers the benefits and strength of integrated management to reduce CRD. Under field conditions, the biological treatments provided crop protection comparable to conventional methods (chemical dressings and Azoxystrobin applications throughout the growing season). Even more so, the bio-friendly seed coating provides a relatively economically feasible solution that can fit any cultivation method and be integrated with various agricultural practices. The same protocol with some adjustments can include different eco-friendly beneficial bacterial or fungal species and can be combined with varying chemical fungicides [55]. Indeed, when administered as seed treatments, soil treatments, or foliar sprays, bio-control agents, including *Trichoderma* sp., *Bacillus subtilis*, and *Pseudomonas* sp., are valuable in controlling diseases and promoting crop growth [32]. A careful pre-examination of those combinations is needed to verify no preparation toxicity toward the bio-protective species [31].

Moreover, bioagents and chemical compounds (type and dosage) should be safe for the plants. An example of a harmful imbalanced treatment was demonstrated here where the combination of Azoxystrobin sprinkling and irrigation (especially if the seeds were also pre-treated chemically) led to growth suppression and CRD-enhanced burst. Such a consequence is most likely the result of phytotoxic overdose. Intriguingly, in the biological seed-coating plants, the double Azoxystrobin treatment (sprinkling and irrigation at low dosage, Sp + D200) was significantly less harmful (*p* << 0.05, Figure 2). It is possible that despite the toxic stress that impaired the plant’s immunity to CRD, the biofriendly *Trichoderma* species managed to prevent the pathogen proliferation. This case needs more investigation, but it highlights the bio-protective suite value. 

Thus, what are the future stages in using this information to the advantage of farmers and markets? First, it will be best to test the successful bio-based integrated protocols in other cotton fields infected by *M. phaseolina* to verify the treatments’ stability and strength under various agricultural practices, soil and climate conditions, and pathogen strains. 

Second, the successful control management, based on *Trichoderma* spp. mix seed coating combined with low-dosage Azoxystrobin sprinkling with seeding or irrigation at two intervals throughout the season, should be optimized. Such adjustments include testing different *Trichoderma* species in the seed-coating formula. Aly et al. (2007) [54] showed that a specific *Trichoderma* isolate could be highly effective against a single *M. phaseolina* strain while having no effect on other isolates of the same fungus. Therefore, evaluating the antagonist’s species against as many pathogen strains as feasible is necessary. This method increases the chances of finding new efficient bio-agents against different *M. phaseolina* strains. On the chemical side of the treatment, adjusting the timing and number of applications of the Azoxystrobin irrigation throughout the season may be crucial [56]. Also, decreasing the fungicide dosages to the minimum while keeping the treatment efficient will be informative and valuable. 

It was formerly reported that applying Azoxystrobin (Mirador 250 SC) via a drip irrigation line positioned between the rows was ineffective in terms of growth promotion and CRD symptom elimination [9]. This lack of significance is explained by the fact that the chemical was dispersed by the irrigation extension, which was placed between the rows, away from the plants, and prevented the chemical from reaching sufficient effective concentration inside the plants. An additional explanation is that such irrigation causes soil dehydration, expansion, and cracking of the tread paths between the plots, leading to plant phelloderm injury and facilitating *M. phaseolina* root colonization [35]. Moreover, higher soil temperatures during the hot summer increased drought stress and worsened this problem, resulting in more fissures [9], making it easier for *M. phaseolina* plants’ root infiltration and establishment [4].

In light of these conclusions, in the current study, we placed drip lines on the bare soil tread paths between the rows to maintain uniformity in soil moisture, improve fungicide movement in the ground, and reduce soil cracking and damage to the roots. Indeed, this practice noticeably impacted the plants’ development, yield, and health in the low dosage of Azoxystrobin irrigation treatments. 

Another essential aspect that should be considered is the high risk of emerging Azoxystrobin-resistant pathogen strains, primarily if such management operates on a large scale over a few years [57,58,59]. Indeed, agro-pesticides with a sole target site are at constant risk of fungicide resistance [60]. Azoxystrobin and other strobilurins prevent mitochondrial respiration. It is the quinone outside inhibitor (QoI) fungicides that inhibit fungi growth by attaching to the quinol oxidation (Qo) site of the electron transport chain’s mitochondrial cytochrome bc1 complex and block the synthesis of ATP [61]. More than 20 genera of fungal diseases, including *Rhizoctonia solani*, *Alternaria alternata*, *Botrytis cinerea*, *Venturia inaequalis*, and *Mycosphaerella graminicola*, have been shown to harbor QoI fungicide resistance mutations [62]. While the information regarding such genetic changes in *M. phaseolina* is scarce, such potential exists, and thus, implementing fungicide resistance management measures is critical [62]. More potent chemical agents should be studied for the integrated control to increase our possibilities for limiting *M. phaseolina* from developing fungicide resistance capability [16,45]. Also, incorporating two or more active ingredients with a different mode of operation is necessary for long-term use to reduce fungicide resistance [63,64]. 

Remote aerial imaging of crop canopy visible light colors (RGB), vegetation vigor degree (GRVI), and stress/dehydration (temperature) is a powerful research tool for detecting delicate variations in the plants’ health and overall field condition [42]. Such potential was demonstrated in the current work, encouraging the expansion of this tool’s usage and maximizing its potential in such an area of study. Advanced equipment and artificial intelligence tools push this technology forward to greater sensitivity and capabilities [65,66]. The diverse applications of this high-end technology already include precision irrigation management [42], disease early detection [67], patterns of disease appearance in the field, studying aggressive fungal strains’ impact [68], evaluating prevention practices, and assessing crop resistance [69]. These uses and new applications yet to be developed provide powerful tools for phytopathologists in plant disease management.

Field crops are at constant exposure and risk to other challenging colonizers in the field soil microbiome. While cotton is susceptible to numerous pathogens and pests, biotechnological tools are promising solutions for preventing losses, minimizing production costs, and providing ecologically friendly control systems for the market and global production [1]. The current research results align with this trend and provide solid intervention for safeguarding the crops in high CRD-risk areas. Advances in cotton genome sequencing, editing, and improved transgenic technology are set to produce new stress-resistant cotton varieties [70]. Integrating these with effective bioagents could revolutionize phytopathogen sanitation, reducing losses and benefiting the environment.

As global warming continues, more and more cotton cultivation fields can become vulnerable to potential *M. phaseolina* harm. As a result, the disease’s natural range will presumably grow, and more severe outbreaks may occur. Thus, it is imperative to discover and enhance efficient control methods to safeguard cotton crops. Future research on the effects of the control strategies described here on additional cotton soil diseases and other commercial field plants is likewise an interesting and worthwhile endeavor. Meanwhile, it is essential to highlight the challenges posed by the absence of readily available biopesticides in the commercial market [71]. Issues such as the instability of *Trichoderma* spp. in dynamic environments and the difficulties in managing their use, including the limited production facilities and supply chain, contribute to the complexity of implementing these solutions in practical applications.

## 5. Conclusions

Charcoal rot (CRD) of cotton is a devastating disease with few management options. The disease causal agent, *Macrophomina phaseolina* (Tassi) Goid, is a soil-born fungus affecting various significant crops, including cotton. The integrated management of CRD was tested here using biological and chemical pesticides in a growth room and at a commercial scale. The bio-seed coating with *Trichoderma* spp., followed by the application of the fungicide Azoxystrobin, was tested in comparison to only biological seed coating or a completely chemical approach, based on only seed coating or in combination with Azoxystrobin addon along the season. In the growth room trial, comparing the bio-based coating treatments versus their parallel chemical coating partners in most cases resulted in similar outcomes on both sampling dates. In contrast, negative effects were highlighted on plant development when applying the hardest chemical approach based on seed coating followed by repeated fungicide application. In the field, the integrated approach is not performed differently than the chemical approach, considering plant symptoms and root infection. Some integrated treatments appeared promising in reducing CRD symptoms, like a biochemical coating combined with Azoxystrobin sprinkling at sowing. A significant advantage to the plants’ health in the field was found in the bio-chemo integrated management over the chemical intervention (considering all treatments in each category together). Regarding yield, no differences were measured among the bio- and chemo-based treatments. This could even mean that the seed coating only with *Trichoderma* provides enough protection to sustain yield and control CRD as well as all the other treatments differently combined. Still, such a conclusion should be verified by repeating the field trial. The findings of this research propose an environmentally friendly method for controlling *M. phaseolina* in cotton fields. Combining biological and chemical intervention capacity to protect cotton crops from CRD is considered eco-friendly since it can significantly reduce chemical pesticides while enhancing and stabilizing the bio-agents’ ability to eliminate the pathogen under the fields’ unstable environment. The results of the current study encourage future adjustments and implementation of the control strategies described here on additional cotton soil diseases and other crops.

## Figures and Tables

**Figure 1 jof-10-00250-f001:**
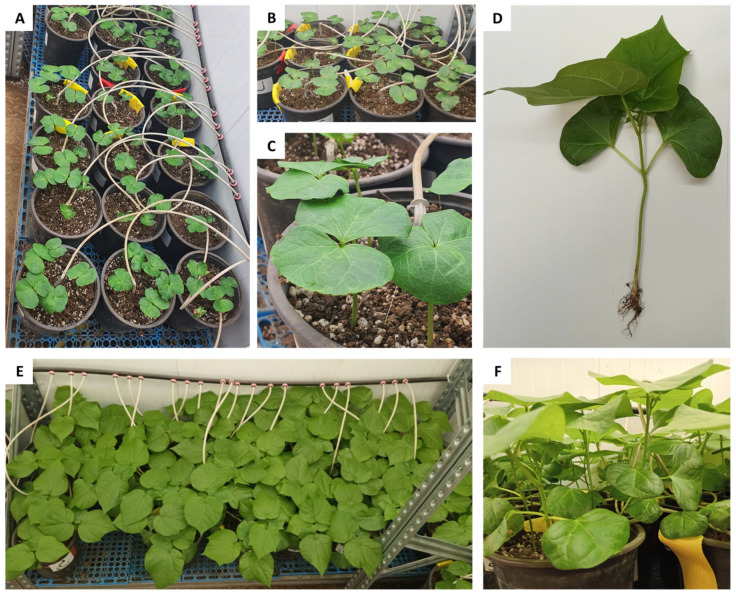
Pot experiment. Thinning from five to one plant per pot on day 29 post-sowing (**A**–**C**). A representative well-developed plant on day 29 post-sowing (**D**). Plants at harvest on day 52 post-sowing (**E**,**F**).

**Figure 2 jof-10-00250-f002:**
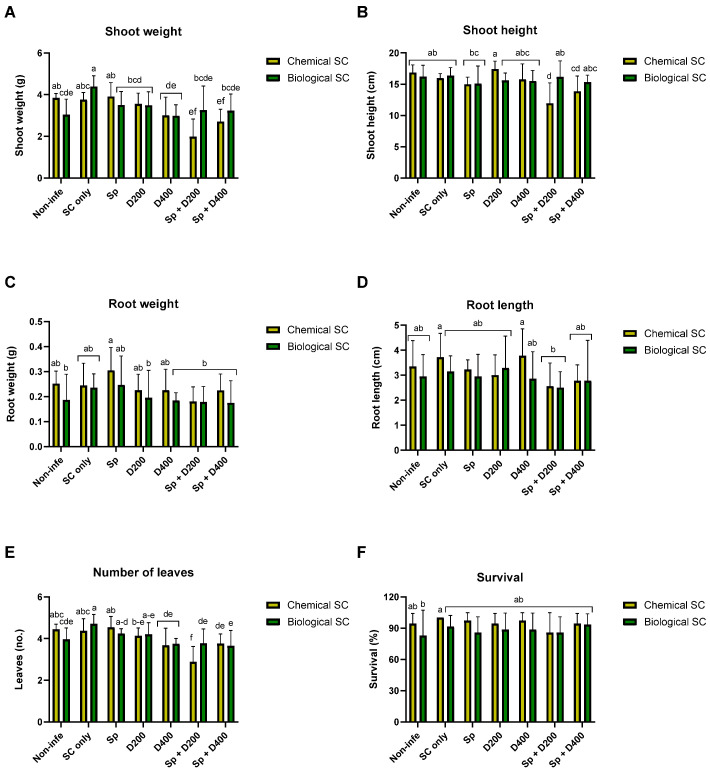
Growth parameters and survival rate of the plants after the different treatments in the pot assay on day 29. Shoot fresh weight (**A**), shoot height (**B**), root weight (**C**), root length (**D**), number of leaves (**E**), and survival rate (**F**). The treatments and the control (Non-infe) were performed once with chemical seed coating (Chemical SC) and once with bio-coating (*Trichoderma* species mix, Biological SC). The treatments are seed coating solely (SC only), seed coating with Azoxystrobin sprinkling (Sp) or irrigation at low (D200) or high (D400) dosage, and a combination of the three (Sp + D200 and Sp + D400). The control mock were healthy, non-infected plants that underwent solely biological or chemical coating. The experiment included 6–7 biological repetitions (average/pot per treatment). Error lines represent the standard error. The statistical significance of variance was assessed using the two-way analysis of variance (ANOVA) and the post-hoc Fisher’s least significant difference (LSD) test. Statistical significance is shown by different letters above the chart bars (a–f).

**Figure 3 jof-10-00250-f003:**
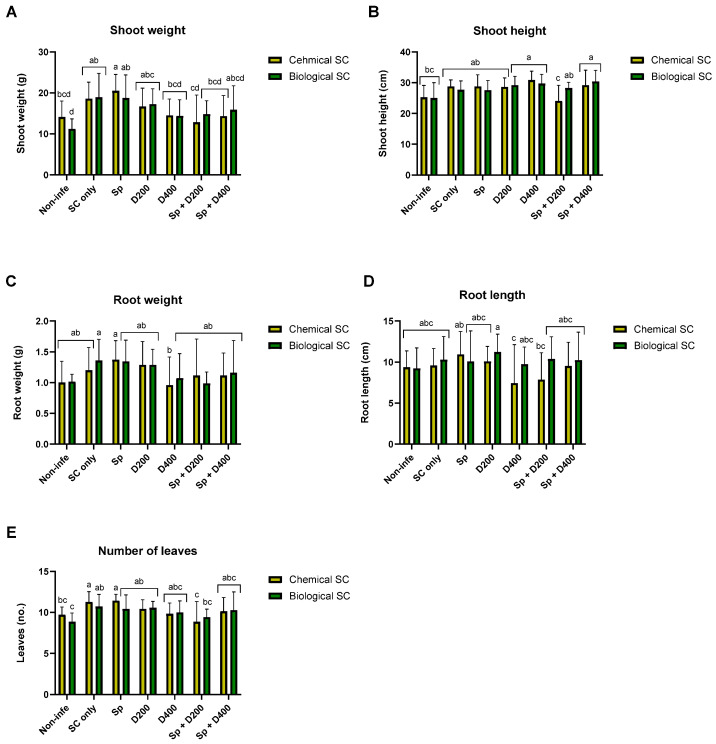
Growth parameters of the plants after the different treatments in the pot assay on day 52. Shoot wet weight (**A**), shoot height (**B**), root weight (**C**), root length (**D**), and number of leaves (**E**). The treatments and the control (Non-infe) were performed once with chemical seed coating (Chemical SC) and once with bio-coating (*Trichoderma* species mix, Biological SC). The treatments are seed coating solely (SC only), seed coating with Azoxystrobin sprinkling (Sp) or irrigation at low (D200) or high (D400) dosage, and a combination of the three (Sp + D200 and Sp + D400). The control mock was healthy, non-infected plants that underwent solely biological or chemical coating. The experiment included seven biological repetitions (average/pot per treatment). Error bars represent a standard error. The statistical significance (set by two-way ANOVA) is symbolized by different letters above the chart bars (a–d).

**Figure 4 jof-10-00250-f004:**
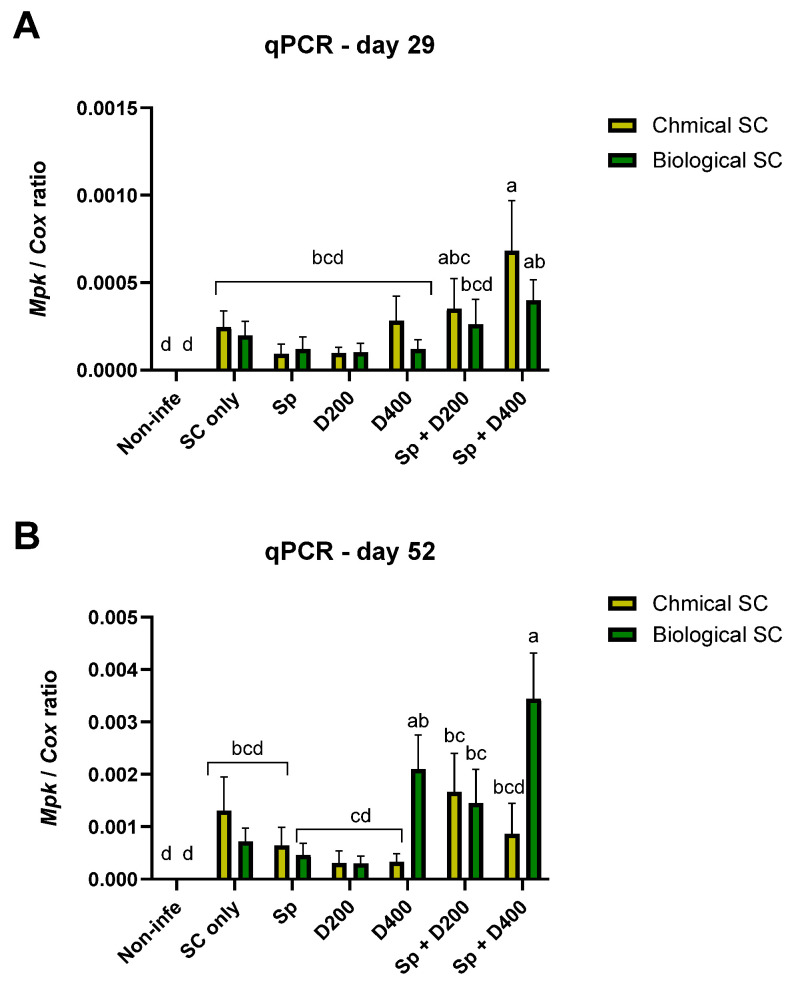
Variation of *Macrophomina phaseolina* DNA in the root tissue of plants sampled on day 29 (**A**) and 52 (**B**) post-sowing after the different treatments in the pot experiment. The treatments and the control (Non-infe) were performed once with chemical seed coating (Chemical SC) and once with bio-coating (*Trichoderma* species mix, Biological SC). The treatments are seed coating solely (SC only), seed coating with Azoxystrobin sprinkling (Sp) or irrigation at low (D200) or high (D400) dosage, and a combination of the three (Sp + D200 and Sp + D400). The control mock comprised healthy, non-infected plants that underwent solely biological or chemical coating. The Y-axis displays the ratio of the specific *M. phaseolina* DNA to the housekeeping gene-encoding cytochrome C oxidase (COX). In most treatments, values represent an average of 7 (day 29) or 4–5 (day 52) plants per treatment. Error lines represent a standard error. The statistical significance of variance was tested using the two-way ANOVA assay, which is shown by different letters above the chart bars (a–d).

**Figure 5 jof-10-00250-f005:**
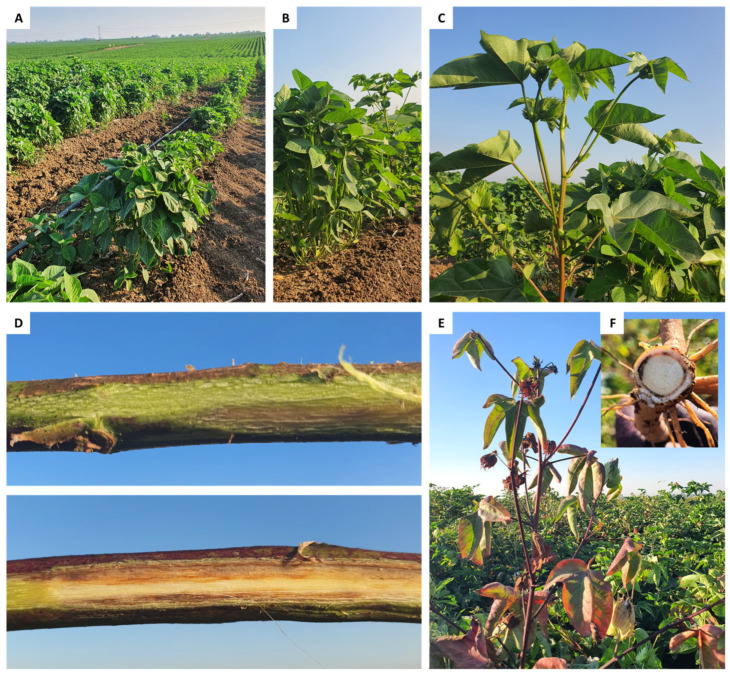
Commercial field experiment. Plant at midseason, 74 days post-sowing (**A**–**C**). Longitudinal section of a healthy (**D**, upper) and *Macrophomina phaseolina* infected (**D**, lower) plant stem. Symptoms of plant wilting (**E**). Cross-section of a diseased stem (**F**).

**Figure 6 jof-10-00250-f006:**
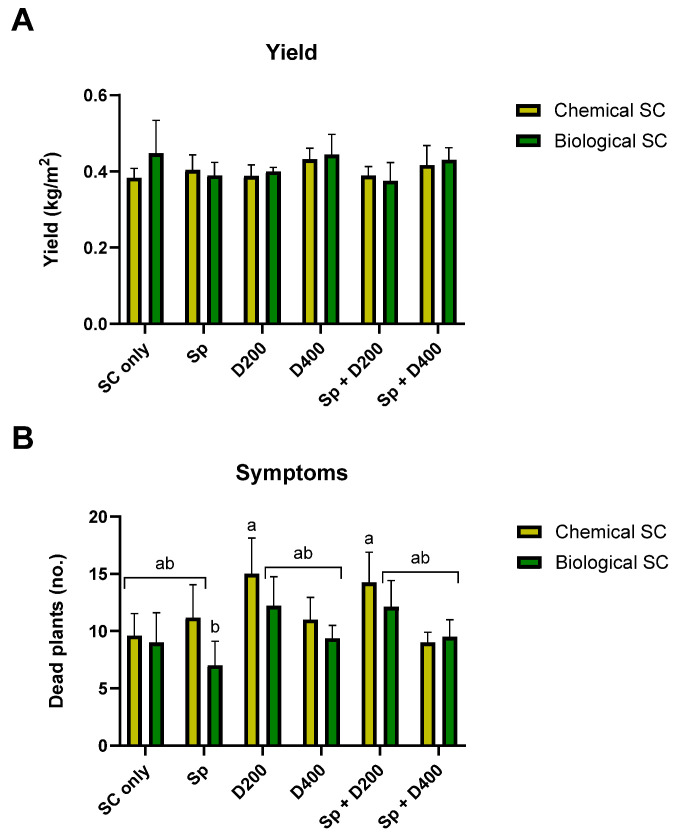
The yield valuation (**A**, day 210) and symptoms assessment (**B**, day 151) in the commercial field trial. The treatments and the control (Non-infe) were performed once with chemical seed coating (Chemical SC) and once with bio-coating (*Trichoderma* species mix, Biological SC). The treatments are seed coating solely (SC only), seed coating with Azoxystrobin sprinkling (Sp) or irrigation at low (D200) or high (D400) dosage, and a combination of the three (Sp + D200 and Sp + D400). Each treatment group and the control group were replicated 3–6 times. Error bars represent a standard error. The statistical significance of variance was tested using the two-way ANOVA assay and, when existing (**B**), is signified by different letters above the chart bars (a,b).

**Figure 7 jof-10-00250-f007:**
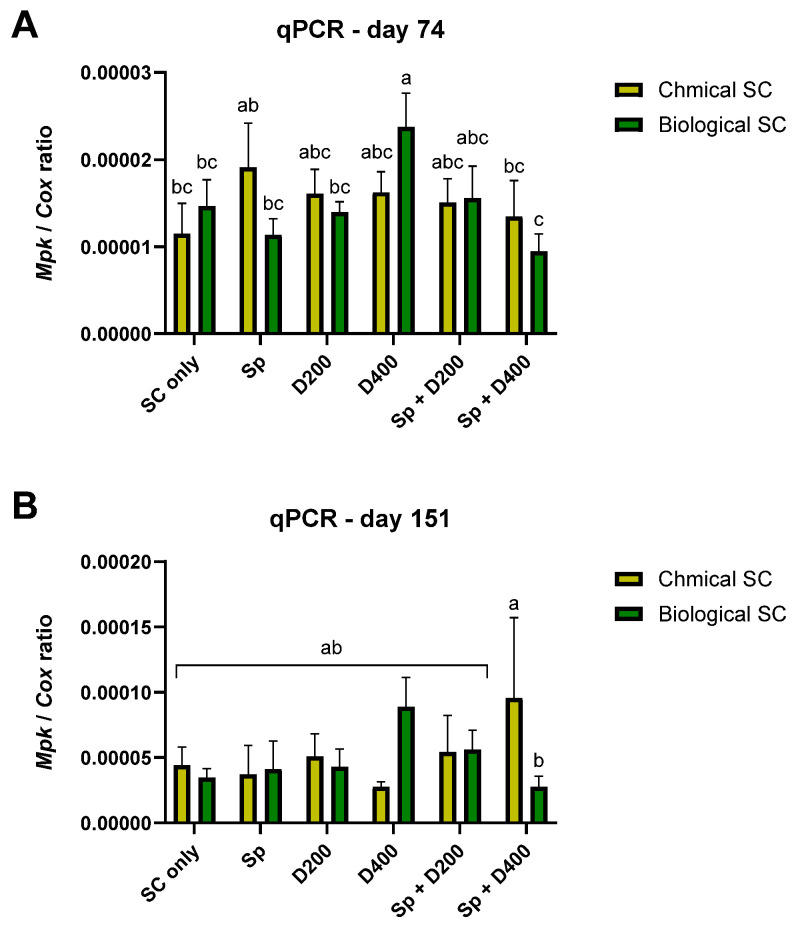
Variation of *Macrophomina phaseolina* DNA in the root tissue of plants sampled on day 74 (**A**) and 151 (**B**) post-sowing after the different treatments in the commercial field trial. The treatments and the control (Non-infe) were performed once with chemical seed coating (Chemical SC) and once with bio-coating (*Trichoderma* species mix, Biological SC). The treatments are seed coating solely (SC only), seed coating with Azoxystrobin sprinkling (Sp) or irrigation at low (D200) or high (D400) dosage, and a combination of the three (Sp + D200 and Sp + D400). The Y-axis presents the ratio of the specific *M. phaseolina* DNA to the COX housekeeping gene. In most treatments, values represent an average of 6–8 repetitions (plants per treatment). In a few instances, 4–5 repetitions per treatment were due to outlier substruction. Error lines represent a standard error. The statistical significance was tested using the two-way ANOVA assay and is signified by different letters above the chart bars (a–c).

**Figure 8 jof-10-00250-f008:**
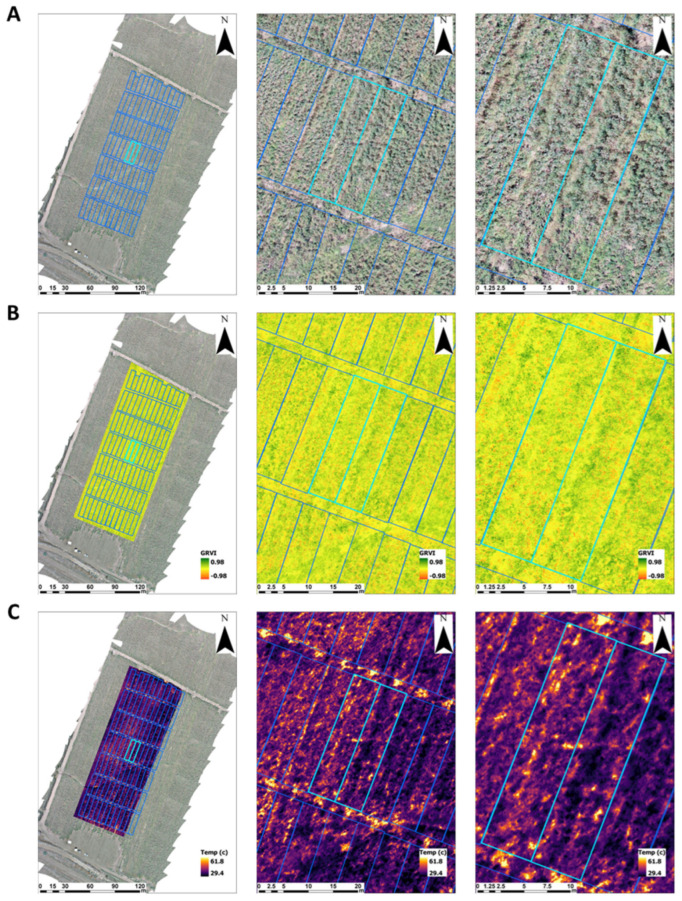
Aerial imaging of the commercial field trial at 162 days post-sowing, obtained by aerial remote sensing: high-resolution visible range (RGB, **A**), green–red vegetation index (GRVI, **B**), and thermal imaging (**C**). Two adjacent representative plots in the middle of the field are shown: plot 44 on the left treated with chemical seed coating solely (SC only), and plot 53 on the right, treated with *Trichoderma* spp. mix seed coating + Azoxystrobin sprinkling and irrigation at low dose treatment (Sp + D200).

**Table 1 jof-10-00250-t001:** *Trichoderma* species isolates that were employed in this research.

Species	Designation	Origin	Reference
*Trichoderma* sp. O.Y. 7107	T7107	*Psammocinia* sp. ^1^	[30,34]
*Trichoderma longibrachiatum*	T7407	*Psammocinia* sp. ^1^	[20,30,34]
*Trichoderma asperellum*	P1	*Zea mays*, Prelude cv.	[14,30]

^1^ Mediterranean sponge *Psammocinia* sp.

**Table 2 jof-10-00250-t002:** The trial architecture of the pot assay in the growth room: description of the different treatments, fungicide dosage, and application time.

No.	Treatment	Designation	Azoxystrobin Dosage (Active Ingredient)	Timetable
1	Chemical seed coating ^1^	SC only	Non (Control)	Before seeding
2	Biological seed coating with *Trichoderma* species mix ^2^			
3	Chemical seed coating and AS ^3^ sprinkling in the sowing hole.	Sp	0.88 mg dissolved in 20 mL DDW ^4^ (equivalent to 200 mL/0.1 ha)	With the seeding
4	Biological seed coating and AS sprinkling in the sowing hole.		0.44 mg dissolved in 20 mL DDW (equivalent to 100 mL/0.1 ha)	
5	Chemical seed coating and AS irrigation (low dosage)	D200	0.88 mg dissolved in 10 mL DDW	10- and 21-days post sowing
6	Biological seed coating and AS irrigation (low dosage)			
7	Chemical seed coating and AS irrigation (high dosage)	D400	1.76 mg dissolved in 10 mL DDW (equivalent to 400 mL/0.1 ha)	
8	Biological seed coating and AS irrigation (high dosage)			
9	Chemical seed coating, AS sprinkling, and irrigation (low dosage)	Sp + D200	As in treatments 3–4 + 5–6	With the seeding and 10- and 21-day post sowing
10	Biological seed coating, AS sprinkling, and irrigation (low dosage)			
11	Chemical seed coating, AS sprinkling, and irrigation (high dosage)	Sp + D400	As in treatments 3–4 + 7–8	
12	Biological seed coating, AS sprinkling, and irrigation (high dosage)			

^1^ standard mix of thiram, captan, carboxin, and Metalaxyl-M; ^2^ mix of fungal spores, mycelium fragments, and extrolites from Trichoderma T7107, T7407, P1 cultures; ^3^ AS: Azoxystrobin; ^4^ DDW: double-distilled water.

**Table 3 jof-10-00250-t003:** The growth room pot experiment’s dates.

**Date**	**Inoculation and Sowing**	**Days from Sowing**
7 May 2023	1st inoculation (sterilized infected millet grains)	−7
14 May 2023	Seeding and pesticide (Azoxystrobin) sprinkling	0
21 May 2023	2nd inoculation (3 discs/sprout)	7
28 May 2023	3rd inoculation (3 discs/sprout)	14
**Date**	**Pesticide Irrigation Treatments and** **Above-Ground Sprouting Assessment**	**Days from Sowing**
24 May 2023	Pesticide I application and soil surface peek evaluation	10
4 June 2023	Pesticide II application (11 days from Pesticide I)	21
**Date**	**Sampling**	**Days from Sowing**
24 May 2023	Above-ground emergence estimation	10
12 June 2023	Mid-experiment sampling and thinning	29
5 July 2023	Final sampling	52

**Table 4 jof-10-00250-t004:** The commercial field experiment’s dates.

Date	Seeding and Sprouting Assessment	Days from Sowing
9 April 2023	Seeding	0
23 April 2023	Soil surface peek valuation	14
	**Irrigation and Azoxystrobin treatments**	
22 May 2023	Watering opening	43
23 May 2023	1st Azoxystrobin irrigation	44
26 June 2023	2nd Azoxystrobin irrigation	78
	**Sampling and harvest**	
22 June 2023	Midseason sampling	74
7 September 2023	End season sampling	151
18 September 2023	Remote sensing (visible and thermal imaging)	162
5 November 2023	Harvest and yield assessment	210

**Table 5 jof-10-00250-t005:** Primers used for quantitative Real-Time PCR *Macrophomina phaseolina* tracking.

Pairs	Primer	Sequence	Uses	Amplification	References
Pair 1	MpKFI MpKRI	5′-CCGCCAGAGGACTATCAAAC-3′ 5′-CGTCCGAAGCGAGGTGTATT-3′	Target gene	300–400 bp *M. phaseolina* species-specific fragment	[40]
Pair 2	COX-F COX-R	5′-GTATGCCACGTCGCATTCCAGA-3′ 5′-CAACTACGGATATATAAGRRCCRRAACTG-3′	Control	Cytochrome c oxidase (COX) gene product	[37,41]

**Table 6 jof-10-00250-t006:** Assessment of the aerial GRVI index and the thermal imaging data after the different treatments in the field trial, expressed as a percentage of the control—chemically (Captan) coated seeds plants ^1^.

		GRVI Index			Thermal Imaging		
	Treatment	Mean	SE	%	Mean	SE	%
Chemical SC ^2^	SC only	0.0928	0.0548	100%	35.33	1.277	100.00%
	Sp	0.0951	0.056	103%	34.96	1.212	98.90%
	D200	0.0929	0.0531	100%	35.69	1.288	101.00%
	D400	0.0912	0.0532	98%	35.36	1.259	100.10%
	Sp + D200	0.0961	0.0535	104%	35.29	1.246	99.90%
	Sp + D400	0.0932	0.0554	100%	35.27	1.347	99.80%
Biological SC	SC only	0.0922	0.0547	99%	35.19	1.234	99.60%
	Sp	0.0899	0.0517	97%	35.63	1.269	100.80%
	D200	0.0951	0.0565	102%	35.54	1.342	100.60%
	D400	0.0926	0.0538	100%	35.23	1.265	99.70%
	Sp + D200	0.0917	0.0519	99%	35.59	1.318	100.70%
	Sp + D400	0.0943	0.0553	102%	35.1	1.297	99.30%

^1^ The treatments were performed once with chemical seed coating (Chemical SC) and once with bio-coating (*Trichoderma* species mix, Biological SC). The treatments are seed coating solely (SC only), seed coating with Azoxystrobin sprinkling (Sp) or irrigation at low (D200) or high (D400) dosage, and a combination of the three (Sp + D200 and Sp + D400). The best and the least influential treatments in controlling the CRD fungus, *M. phaseolina,* are highlighted in green and pink, respectively. ^2^ SC—seed coating.

## Data Availability

The datasets generated and/or analyzed during the current study are available from the corresponding author upon reasonable request.

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
