# Peer review of "Integrated Management of the Cotton Charcoal Rot Disease Using Biological Agents and Chemical Pesticides"

_jof, 2024, doi:10.3390/jof10040250_

Round 1
Reviewer 1 Report
Comments and Suggestions for Authors
Overall Comments:
This research aims to build, improve and deepen our understanding of novel approaches combining biological and chemical pesticides to provide data supporting the stability and leading edge of eco-friendly bioprotective Trichoderma species. The study design included sprouts in a growth room and commercial field plants receiving the same treatments. The data is extensive, the research findings are significant, and the language is well-written. It is recommended for acceptance after revisions.
Specific Comments:
1. Regarding "biological and chemical management practices" in Table 2, it is suggested to provide a more detailed breakdown of the experimental variables. Clarify which fall under biological treatment and which fall under chemical treatment. Additionally, it is recommended to include illustrative diagrams of the treatments for better clarity. From my understanding, seed coating involves both chemical agent coating and biofungicide coating, followed by different fungicidal chemical treatments. The actual control methods involve chemical control and a combination of biological and chemical control.
2. For Figures 2 and 3, as well as Figures 6, and Tables 7 and 8, where you differentiate between chemical and biological methods, I find it less precise. It is recommended to revise it to specify chemical and biological methods for seed coating.
3. Rectify the writing error in the labeling of "chemical" in Figures 2 and 3. In addition, please note that the "L" in the unit “mL”and “μL” in Table 2 and the main text must be capitalized.
4. If possible, convert remote sensing image results into numerical data for comparative analysis: Translate remote sensing image results into numerical values and perform a comparative analysis to enhance persuasiveness. These modifications aim to improve the clarity, accuracy, and persuasiveness of the study.
Author Response
Responses to Reviewer 1's comments
We thank the reviewer for investing substantial effort, which undoubtedly contributed to this manuscript. The remarks and suggestions improved this paper's scientific soundness and accuracy. Your contribution is greatly appreciated.
This research aims to build, improve, and deepen our understanding of novel approaches combining biological and chemical pesticides to provide data supporting the stability and leading edge of eco-friendly bioprotective Trichoderma species. The study design included sprouts in a growth room and commercial field plants receiving the same treatments. The data is extensive, the research findings are significant, and the language is well-written. It is recommended for acceptance after revisions.
Reply: Thank you for the positive evaluation of our manuscript. All your remarks and suggestions were addressed carefully and thoroughly, as detailed below.
Specific Comments:
- Regarding "biological and chemical management practices" in Table 2, it is suggested to provide a more detailed breakdown of the experimental variables. Clarify which falls under biological treatment and which falls under chemical treatment.
Reply: Thank you for this important remark. Table 2 was modified according to your advice to describe all treatments applied fully.
- Additionally, it is recommended to include illustrative diagrams of the treatments for better clarity. From my understanding, seed coating involves both chemical agent coating and biofungicide coating, followed by different fungicidal chemical treatments. The actual control methods involve chemical control and a combination of biological and chemical control.
Reply: You are right. An illustrative diagram of the treatments was added as suggested. The new figure, Figure 1, summarizes the pots' assay experimental design scheme. The same protocol (with adjustments) was applied in the field.
- For Figures 2 and 3, as well as Figures 6, and Tables 7 and 8, where you differentiate between chemical and biological methods, I find it less precise. It is recommended to revise it to specify chemical and biological methods for seed coating.
Reply: Thank you for this suggestion. We corrected all the labeling in the graphs and tables to "Biological SC" and "Chemical SC" and added the abbreviation (SC – seed coating) explanation to the text.
- Rectify the writing error in the labeling of "chemical" in Figures 2 and 3.
Reply: We apologize for this typo. The labeling of "chemical" in the figures (now Figures 3 and 4) was corrected.
- In addition, please note that the "L" in the unit "mL" and "μL" in Table 2 and the main text must be capitalized.
Reply: Corrected as advised throughout the text.
- If possible, convert remote sensing image results into numerical data for comparative analysis: Translate remote sensing image results into numerical values and perform a comparative analysis to enhance persuasiveness. These modifications aim to improve the clarity, accuracy, and persuasiveness of the study.
Reply: Thank you for this important advice. According to your suggestion, a new table, Table 10, was added to the text. The table describes the commercial field trial remote sensing data averages of the aerial GRVI index and the thermal imaging data.
Also, the following paragraph was updated (the GRVI index and temperature mean values of the two representative plots were added): "Indeed, Plot 53 (on the right) exhibited more vivid and developed vegetation (Figure 9A), a higher GRVI index (mean of 0.085, Figure 9B), and a lower temperature (mean of 35.5°C, Figure 9C) compared to Plot 44 on the left (mean GRVI index of 0.075 and temperature of 36.6°C)." (lines 591-594).
Reviewer 2 Report
Comments and Suggestions for Authors
This paper provides important validation to the combined biological and chemical intervention potential to shield cotton crops from the charcoal rot disease. The data is sufficient and its analysis is accurate, and this paper will help in the efficient management of the cotton charcoal rot disease.
1. L24, "At the age of 52", it is recommended to change it to "At the age of day 52".
2. Table 1, “Trichoderma sp. O.Y. 7107 (Full italics)” modified to “Trichoderma sp. O.Y. 7107(sp. O.Y. 7107 is non-italicized)”.
3. Figure 2, “Chmical” modified to “Chemical”.
4. Figure citations in the text should be more specific, for example, Figure 2 should be indicated as Figure 2A? Figure 2B? or Figure 2C? and so on.
5. L481-482, “This treatment improved the sprouts' root weight by 21% compared to the non-infected control”, however, I cannot find a significant difference in Figure 2C. The same to “which resulted in a 6% higher survival rate than the mock control group”. Figure 2C and Figure 2F were not analyzed for significant difference? It is recommended to mark significance on the figure.
6. In Table 7, Table 8 and Table 9, it is more reasonable to perform ranking based on the significance of the difference. It is recommended that treatments with insignificant differences be classified into the same rank.
Author Response
Responses to Reviewer 2's comments
We want to express our sincere appreciation to the reviewer for the essential and helpful advice. The time and effort invested are greatly appreciated and certainly contributed to and improved the manuscript. Thank you.
This paper provides important validation to the combined biological and chemical intervention potential to shield cotton crops from charcoal rot disease. The data is sufficient, its analysis is accurate, and this paper will help in the efficient management of the cotton charcoal rot disease.
Reply: Thank you for the positive evaluation of our manuscript. All your remarks and suggestions were addressed carefully and thoroughly, as detailed below.
Comments:
- L24, "At the age of 52", it is recommended to change it to "At the age of day 52".
Reply: Corrected as advised.
- Table 1, "Trichoderma sp. O.Y. 7107(Full italics)" modified to "Trichoderma sp. O.Y. 7107(sp. O.Y. 7107 is non-italicized)".
Reply: Corrected as advised.
- Figure 2, "Chmical" modified to "Chemical".
Reply: We apologize for this typo. The labeling of "chemical" in Figures 2 and 3 (now Figures 3 and 4) was corrected.
- Figure citations in the text should be more specific, for example, Figure 2 should be indicated as Figure 2A? Figure 2B? or Figure 2C? and so on.
Reply: You are right. As suggested, all figure citations were checked thoroughly and corrected to be more specific.
- L481-482, "This treatment improved the sprouts' root weight by 21% compared to the non-infected control." However, I cannot find a significant difference in Figure 2C. The same to "which resulted in a 6% higher survival rate than the mock control group". Figure 2C and Figure 2F were not analyzed for significant differences. It is recommended to mark the significance on the figure.
Reply: Thank you for this important suggestion. We have redone the statistical analysis in Figures 3, 4, and 7, using the two-way analysis of variance (ANOVA) assessment and posterior Fisher's least significant difference (LSD) test. We added letters to describe statistical significance differences between the treatments. The text and the figures' captions were updated accordingly.
- In Table 7, Table 8, and Table 9, it is more reasonable to perform a ranking based on the significance of the difference. It is recommended that treatments with insignificant differences be classified into the same rank.
Reply: You are right; the description of the results should focus on those that gained significant differences from the control. It should be noted that effective treatments for growth promotion, health, and yield do not consistently achieve statistical significance due to high variability (standard errors). Such fluctuations are predicted in the growth room due to the objective challenge of gaining uniform infection and disease. Those variations are magnified under the unstable environment of the field assay and the nonuniform spread of the disease in the soil. So, we compare the averages of the treatments to the control (as done in Tables 7, 8, and 9) to predict their performance better. Regarding the ranking, as you and reviewer 3 suggested, we eliminated the Rank columns from all Tables, focusing the analysis on the average comparison.
Reviewer 3 Report
I appreciated very much the experiment in open field because too often the studies remain limited to laboratory or greenhouse conditions. However, the manuscript needs major revision to be worth publication.
Some general considerations:
First of all, I found the manuscript too long, starting from the Introduction up to the Conclusion. The author should shorten all the chapters.
In the Introduction, avoid discussing the literature in detail, and do that in the Discussion chapter, instead.
The Materials and Methods chapter should be checked also for repetitions.
The Results are presented in a very confusing way. The reader cannot catch the main results (that are pointed out in the conclusion section). Sometimes the statistical significance is reported in the Figures by asterisks, sometimes by letters, and sometimes is absent. The authors should present the results based on the statistical analysis, at first. The ranking has no meaning. At maximum the best and the worst treatment can be pointed out, (if they are significantly different, of course).
The Figure/table captions need to be completely rewritten (as suggested below)
The Discussion should put the results in relation to the existing literature, avoiding referring mainly to the previous experiment conducted by the same authors.
The Conclusion should not repeat the entire story, and the results, but focus on the importance of the results, the innovation, possible utilization, etc.
The title: add ‘agents’ after biological
Line 24, add days after 52
Line 34 and 35 delete fungus, insert Macrophomina phaseolina
Line 161 add “integrated “ before “pest management approach”
Line 165 add “enhanced” or a similar verb before “health indexes”
Line 179 Azoxystrobin is a systemic fungicide, not an elicitor. Please, correct
Line 183 at sowing or by irrigation
Line 185 targeting the DNA of the CRD agent, M. phaseolina,
Line 190-210 Rationale. This paragraph should be eliminated or shortened and eventually integrated at the end of the Introduction. The Introduction also is too long and should be shortened.
Line 212 M. phaseolina
Line 216 “All fungal species”. Only one species has been cited (M, phaseolina). If this technique refers also to Trichoderma, the information about Trichoderma should be anticipated here
Line 230-231 Please delete, since it is fully described in paragraph 2.3.3
Line 250 use ml instead of cc, according to the International standards
Line 278 complementary
Line 303 and throughout the text. What do you mean by The above ground peeking percentage? The emergence percentage? Please correct like this.
Line 318 substitute “impacted by” with “due to”
Line 321 what is “a garden bed”?
Line 322-324 and 335Please use m instead of meters, l or L instead of liters, i. e. use the International standard for Unit measurements. Dunam is not a standard measurement unit. Convert it into ha.
Line 325-326 It was already described at 2.3.1 paragraph. Please avoid repetitions
Line 350 please reword the sentence
Line 353 move this Table among Supplementary materials
Line 372 and 373 Correct into 106
Line 374-375 What about the sharp pH decrease in the residual liquid? Are you sure?
Line 386 The last step should be with sterile distilled water. Please invert the two steps of washing
Line 388 To extract the DNA
Line 388-389 (CTAB) buffer solution
Line 391 min
Line 420, 450 use the acronym only
Line 445 posterior is not the correct term
Line 451 M. phaseolina
Line 448-460 Please show the results without repeating all this
Line 463-473 Please do not repeat all the experimental design. Just give a brief description of the images
Line 474-475 ‘Above-ground peeking percentage estimation, conducted ten days from sowing, showed values between 89% and 100% sprouting’ should be correct into ‘Ten days from sowing, the emergence percentage was between 89% and 100%’
Line 478-479 delete the explanation of the treatment in parenthesis
Line 480-481 delete “in these regulated terms”
Line 488 parameters instead of indices
Line 503. Correct the title of the Figure 2 into Growth parameters and survival index of the plants in the pot experiment, measured on day 29. Check the list of graphs because it does not correspond to the graphs in the Figure. Shorten as much as possible the Figure legend, without repeating the methods. Just explain all the acronyms, the names of the treatments used, and symbols significance. Explain what does it mean “chemical” or “biological” as they appear in the legend of each graph. Delete day 29 from the title of each graph. The Dunnett test should be recalled also in the Statistical analysis section. The representation of the statistical significance is not immediately clear. Please show which treatment is significantly different from the other in a clearer way, choosing only one significance level, by the use of letters, as done in Figure 4.
The same for Figure 3. The title could be Growth parameters and survival index of the plants in the pot experiment, measured on day 52.
Line 534 -540 and table 7. I disagree about making averages among the percentages over the control of the different parameters. The best and the worst treatments could be identified as the ones that obtained the highest number of green or pink labels at each sampling date, with a major focus on the final date, which is the most interesting. The other should not be ranked. Thus eliminate the last column “Total” and the subcolumns Average and Rank. Adjust the main text and the legend accordingly.
Figure 4. Change the title of the Figure, according to what it shows. Shorten the legend. Explain the treatment and the symbols, without recalling other Figures. Each figure must be self-explanatory. Avoid repeating methods details. Add the statistical test used for comparisons. See the comments already given for the other Figures
Table 8. delete completely. At a maximum, show the results of PCR only, for each sampling day without average or ranking.
Adjust the main text and the legend, accordingly
Figure 5. Shorten the caption, as already pointed out for Figure 1. Avoid methods description etc…
Figure 6 The statistical analysis results should be reported. Add the letters to indicate statistical significance. Adjust the Figure caption as already pointed out for the other Figures
Line 583 dispersal is not the right term
Line 607-609 How can you say that biocoating gave advantage in yield and symptoms? Is it statistically supported? I do not see any statistical lettering in figure 6. Add letters and adjust the comments. Avoid to base this statement only on ranking (generally avoid ranking). Based on the Figure 6, yield appears stable, neither affected by the chemical or biological coating, nor by the treatment!
Symptoms: I am not sure that the biological approach is better than the chemical one, looking at the graph. Add statistics.
Line 612 interphase is not the right term
Figure 7 Rephrase the title and the caption, avoid referring to other Figures, explain the name of the treatments, avoid method description etc..
Table 9 Again, restructure this table, as already suggested, without total, without averaging and without ranking
Line 681 milestone appears an excessive term

Author Response
Responses to Reviewer 3's comments
We want to express our sincere appreciation to the reviewer for the essential and helpful advice. The time and effort invested are greatly appreciated and certainly contributed to and improved the manuscript. Thank you.
I appreciated very much the experiment in the open field because too often the studies remain limited to laboratory or greenhouse conditions. However, the manuscript needs major revision to be worth publication.
Reply: Thank you for the evaluation of our manuscript. All your remarks and suggestions were addressed carefully and thoroughly, as detailed below.
Does the title describe the article's topic with sufficient precision? It could be more precise like this: Integrated Management of the Cotton Charcoal Rot Disease Using Biological Agents and Chemical Pesticides.
Reply: Right, the title you suggested is indeed more accurate. The title was updated as advised.
Does the Introduction provide a comprehensive yet concise overview about the state of knowledge in the area of research? It is not concise.
Reply: As elaborated below, we carefully and thoroughly re-checked the manuscript and tried our best to simplify the writing and make the text more coherent, clearer, and shorter.
Are the results presented clearly and in sufficient detail, are the conclusions supported by the results and are they put into context within the existing literature? The results are presented confusingly and not appropriately analyzed.
Reply: We followed your suggestion to improve the result section to achieve more clarity and scientific accuracy. The Result chapter was carefully and thoroughly re-checked. We tried our best to simplify the writing, focus the text, and improve the reader’s understanding of the subject matter. See the details below in our reply to your Detail comments.
Some general considerations:
First of all, I found the manuscript too long, starting from the Introduction up to the Conclusion. The author should shorten all the chapters.
Reply: To address this issue, the manuscript was carefully and thoroughly re-checked and edited. We tried simplifying the writing and making the text more coherent, precise, and shorter. See the details below.
In the Introduction, avoid discussing the literature in detail, and do that in the Discussion chapter, instead.
Reply: The Introduction chapter was re-edited and shortened as advice. This editing was made while maintaining the updated scientific knowledge that led to the current work and omitting or moving some information from the Introduction to the Discussion.
The following sentences/paragraphs were deleted from the text:
- "The combined crop exports from these countries amounted to approximately USD 30 billion."
- "Today, there is an agreement among cotton growers in Israel that CRD significantly impacts this cultivar's agriculture and marketing."
- "While the field trial was the first report of CRD bio-friendly treatments on a commercial field in Israel…"
The following sentences/paragraphs were moved to the Discussion:
- "One potentially safe option is to induce systemic resistance in the cotton plants. External agents can trigger a plant's defense mechanism prior to infection, a phenomenon referred to as induced systemic resistance (ISR) [46,47]. Various biotic and abiotic factors have proven effective in inducing ISR in plants as a defense against diverse plant pathogens [48]." (Lines 618-622).
- " As an example for this research line, Trichoderma harzianum (strain SH2303) and a mixture of Difenoconazole-propiconazole were effectively used in tandem to manage the maize southern corn leaf blight agent, Cochliobolus heterostrophus [49]. Additionally, an integrated (bio-chemical) pest control method targeting Magnaporthiopsis maydis in corn [33] has demonstrated the promising potential of this approach. Combining Trichoderma species mixture and low dosage Azoxystrobin reduces disease prevalence and enhances corn crop quantity and quality." (Lines 624-631).
- "In cotton, certain rhizospheric bacteria, particularly those of the Bacillus species, are renowned among biocontrol agents. This reputation is attributed to their ability to produce antimicrobial metabolites, induce systemic resistance, and efficiently establish colonization in soil [50]. A recent study demonstrated the importance of two bacteria strains, Bacillus megaterium ZMR-4 and B. subtilis IAGS-174, along with Benzothiadiazole for inhibition of CRD and growth promotion of cotton under greenhouse and field conditions [51]. In the greenhouse, combining the two bacteria and the chemical pesticides demonstrated 80–83% biocontrol over the pathogen, with individual benzothiadiazole administration showing only 69–71% potential. When used separately, the chosen bacteria suppress the disease in 50–53%. Under field conditions, plants that received a combined application of B. megaterium ZMR-4 and Benzothiadiazole showed the highest biocontrol efficiency (78.9-72.8%) [51]." (Lines 632-643).
The Materials and Methods chapter should also be checked for repetitions.
Reply: We have carefully and thoroughly reviewed the materials and Method section. The section is now better written. Thank you. The changes include:
- Section 2.1 (Rationale of the study design) was deleted from the text, and some of its information was merged into the introduction section.
- Section 2.2 (now section 2.1) was edited and rewritten.
- Table 2 was modified according to Reviewer 1 advice to describe all treatments applied fully.
- An illustrative diagram of the treatments was added as suggested by Reviewer 1. The new figure, Figure 1, summarizes the pots' assay experimental design scheme. The same protocol (with adjustments) was applied in the field.
- The following paragraph was omitted from the text: "…by a mixture of three Trichoderma fungi: T7107, T7407, and P1 (Table 1). This mix was tested alone or in combination with chemical treatments (Table 2) compared with the application of chemical treatments solely."
- The following paragraph was omitted from the text: "Chemical protection was performed by adding Azoxystrobin (commercial formula) according to Table 2. as sprinkling to the sowing fossette (100 or 200 mL to the biologically or chemically coated seeds' pots, respectively). Alternatively, irrigation at 0.88 mg per pot, two times during the cultivation (10 and 21 days from sowing)."
- Two paragraphs (lines 285-302) were deleted and replaced by a short paragraph: "The treatments are summarized in Table 2. Since the standard general pesticide treatment with Captan had no adequate protection against LWD (Ofir Degani personal communication), a double Azoxystrobin sprinkling dosage (200 mL) was applied to the chemical-treated plants. Bio-shielding was carried out in seed coating, as described in [9]. The same bio-coated seeds used in the field assay were used in the growth room trial (see section 2.3.3 for the seeds coating detailed preparation protocol)" (lines 251-256).
- The following sentence was omitted from the text: "The study involved 12 treatments; half were based solely on chemical intervention, and the other half were biological or integrated treatments."
- A few more minor corrections (words and short sentences replacement) were conducted in several places along the text.
The Results are presented in a very confusing way. The reader cannot catch the main results (that are pointed out in the conclusion section).
Reply: as you can see in the revised manuscript, we carefully checked and edited the entire Result section to simplify and clarify the text as much as possible. The changes are reflected in the results’ description and the figure’s legend. All changes were made, paying careful attention to avoid losing any critical data or changing the text’s focus. See the details below in our reply to your specific comments.
Sometimes the statistical significance is reported in the Figures by asterisks, sometimes by letters, and sometimes is absent.
Reply: Thank you for this important suggestion. We have redone the statistical analysis in Figures 3, 4, and 7, using the two-way analysis of variance (ANOVA) assessment and posterior Fisher's least significant difference (LSD) test. We added letters to describe statistical significance differences between the treatments. The text and the figures' captions were updated accordingly.
The authors should present the results based on the statistical analysis, at first. The ranking has no meaning. At maximum the best and the worst treatment can be pointed out, (if they are significantly different, of course).
Reply: You are right; the description of the results should focus on those that gained significant differences from the control. It should be noted that effective treatments for growth promotion, health, and yield do not consistently achieve statistical significance due to high variability (standard errors). Such fluctuations are predicted in the growth room due to the objective challenge of gaining uniform infection and disease. Those variations are magnified under the unstable environment of the field assay and the nonuniform spread of the disease in the soil. So, we compare the averages of the treatments to the control (as done in Tables 7, 8, and 9) to predict their performance better. Regarding the ranking, as you and reviewer 1 suggested, we eliminated the Rank columns from all Tables, focusing the analysis on the average comparison.
The Figure/table captions need to be completely rewritten (as suggested below)
Reply: Based on your advice, all figures and Tables' captions were checked thoroughly and carefully corrected to achieve better clarity and coherence. The captions are now shortened and more focused. See the details below in our reply to your specific comments.
The Discussion should put the results in relation to the existing literature, avoiding referring mainly to the previous experiment conducted by the same authors.
Reply: We improved the Discussion section, as recommended.
The Conclusion should not repeat the entire story, and the results, but focus on the importance of the results, the innovation, possible utilization, etc.
Reply: Thank you for this suggestion. The Conclusion section was edited and improved, and we believe it now focuses more on the importance of the results, the innovation, and possible utilization.
Detail comments
- The title: add 'agents' after biological.
Reply: Right, the title you suggested is indeed more accurate. The title was updated as advised to: Integrated Management of the Cotton Charcoal Rot Disease Using Biological Agents and Chemical Pesticides.
- Line 24, add days after 52.
Reply: Corrected as advised.
- Line 34 and 35 delete fungus, insert Macrophomina phaseolina.
Reply: Corrected as advised.
- Line 161 add "integrated "before "pest management approach".
Reply: Corrected as advised.
- Line 165 add "enhanced" or a similar verb before "health indexes".
Reply: Corrected as advised.
- Line 179 Azoxystrobin is a systemic fungicide, not an elicitor. Please, correct.
Reply: Corrected as advised.
- Line 183 at sowing or by irrigation.
Reply: Corrected as advised.
- Line 185 targeting the DNA of the CRD agent, phaseolina.
Reply: Corrected as advised.
- Line 190-210 Rationale. This paragraph should be eliminated or shortened and eventually integrated at the end of the Introduction.
Reply: Section 2.1. (Rationale of the study design) was omitted from the Materials and Methods section, and some of its information was integrated with the concluding paragraph in the introductory chapter.
- The Introduction is also too long and should be shortened.
Reply: Per your suggestion, we did our best to shorten the introduction section. The changes made to the text are detailed in our reply to your general considerations: comment, no. 2.
- Line 212 phaseolina.
Reply: Corrected as advised.
- Line 216 "All fungal species". Only one species has been cited ( phaseolina). If this technique refers also to Trichoderma, the information about Trichoderma should be anticipated here.
Reply: Indeed, this technique refers also to the Trichoderma species. The text was reorganized according to your advice to clarify this better.
- Line 230-231 Please delete, since it is fully described in paragraph 2.3.3.
Reply: The sentence was deleted according to your recommendation.
- Line 250 use ml instead of cc, according to the International standards.
Reply: Corrected according to your advice throughout the text.
- Line 278 complementary.
Reply: Corrected as per your advice.
- Line 303 and throughout the text. What do you mean by The above ground peeking percentage? The emergence percentage? Please correct like this.
Reply: Corrected throughout the text (in three places).
- Line 318 substitute "impacted by" with "due to".
Reply: The phrase "impacted by" was substituted with "due to"as recommended.
- Line 321 what is "a garden bed"?
Reply: In field crop cultivation, a garden bed refers to a raised area of soil where plants are grown. These beds are typically created to improve drainage, soil aeration, and root growth. The process of creating garden beds involves raising the soil level above the surrounding ground, often using techniques such as plowing, tilling, or hilling. This helps create a well-drained, aerated growing environment that can promote healthier plant growth and increase yields. They can also help facilitate better soil management, weed control, and irrigation efficiency and optimize planting density and spacing, making it easier to manage crops and harvest yields.
A short explanation was added in lines 277-278: "Each experimental plot comprised six rows (in a garden bed, a raised area of soil above the surrounding ground)…"
- Line 322-324 and 335 Please use m instead of meters, l or L instead of liters, i. e. use the International standard for Unit measurements. Dunam is not a standard measurement unit. Convert it into ha.
Reply: Reviewer 1 asked us: "Please note that the "L" in the unit "mL" and "μL" in Table 2 and the main text must be capitalized". So, we will leave the journal editor to determine the preferred way to present the units. We convert "Dunam" into "ha" as suggested.
- Line 325-326 It was already described at 2.3.1 paragraph. Please avoid.
Reply: Right. The sentence was deleted.
- Line 350 please reword the sentence.
Reply: The sentence was rewritten and now reads: "Fertilizers and treatments against various pests were applied throughout the experiment to mitigate risks other than M. phaseolina infection" (lines 305-307).
- Line 353 move this Table among Supplementary materials.
Reply: Table 5 moved to supplementary materials and named Table S1. The text was accordingly updated.
- Line 372 and 373 Correct into 106.
Reply: The typo was corrected to 106 as per your advice.
- Line 374-375 What about the sharp pH decrease in the residual liquid? Are you sure?
Reply: Indeed, this is interesting. We measured this pH drop but didn't notice any influence of this runoff pH on the seeds' vitality. It can be assumed that the dry seeds absorbed the water from the coating solution during their imbibition (swelling) and, in the process, caused an increase in the concentration of the cations in the runoff solution. It also may be the result of materials released from the seeds.
- Line 386 The last step should be with sterile distilled water. Please invert the two steps of washing.
Reply: Indeed. Corrected.
- Line 388 To extract the DNA.
Reply: Corrected as per your advice.
- Line 388-389 (CTAB) buffer solution.
Reply: Corrected as per your advice.
- Line 391 min.
Reply: Corrected throughout the text.
- Line 420, 450 use the acronym only.
Reply: Corrected as advice in those places and a few others.
- Line 445 posterior is not the correct term.
Reply: The term " posterior test" was corrected to "post-hoc analysis."
- Line 451 M. phaseolina.
Reply: Corrected as per your advice.
- Line 448-460 Please show the results without repeating all this.
Reply: Correct. Most of the paragraph was omitted from the text.
- Line 463-473 Please do not repeat all the experimental design. Just give a brief description of the images.
Reply: The experimental design was removed from the figure legend as recommended.
- Line 474-475 'Above-ground peeking percentage estimation, conducted ten days from sowing, showed values between 89% and 100% sprouting' should be correct into 'Ten days from sowing, the emergence percentage was between 89% and 100%'.
Reply: Corrected. Thank you.
- Line 478-479 delete the explanation of the treatment in parenthesis.
Reply: The treatment explanation in parenthesis was deleted according to your advice.
- Line 480-481 delete "in these regulated terms".
Reply: Deleted as suggested.
- Line 488 parameters instead of indices.
Reply: Corrected.
- Line 503. Correct the title of Figure 2 into Growth parameters and survival index of the plants in the pot experiment, measured on day 29.
Reply: Corrected to: "The plants' growth parameters and survival index in the pot experiment on day 29."
- Check the list of graphs because it does not correspond to the graphs in the Figure.
Reply: We checked the graphs list in all figures legends and found no mistakes.
- Shorten as much as possible the Figure legend, without repeating the methods. Just explain all the acronyms, the names of the treatments used, and symbols significance. Explain what does it mean "chemical" or "biological" as they appear in the legend of each graph.
Reply: Thank you. We followed your advice and corrected all figures' legends accordingly. The figures' legends are now shorter and more focused.
- Delete day 29 from the title of each graph.
Reply: The day description in the titles of all graphs in Figures 3 and 4 was deleted.
- The Dunnett test should be recalled also in the Statistical analysis section.
Reply: As detailed in our response in 43, we have redone the statistical analysis in Figures 3 and 4, using the two-way analysis of variance (ANOVA) assessment and posterior Fisher's least significant difference (LSD) test. So, the Dunnett test information was deleted from the text.
- The representation of the statistical significance is not immediately clear. Please show which treatment is significantly different from the other in a clearer way, choosing only one significance level, by the use of letters, as done in Figure 4.
Reply: We have redone the statistical analysis in Figures 3 and 4, using the two-way analysis of variance (ANOVA) assessment and posterior Fisher's least significant difference (LSD) test. As advised, we used letters to describe statistical significance differences between the treatments.
- The same for Figure 3. The title could be Growth parameters and survival index of the plants in the pot experiment, measured on day 52.
Reply: Corrected to: " The plants ' growth parameters and survival index in the pot experiment on day 52."
- Line 534 -540 and table 7. I disagree about making averages among the percentages over the control of the different parameters. The best and the worst treatments could be identified as the ones that obtained the highest number of green or pink labels at each sampling date, with a major focus on the final date, which is the most interesting. The other should not be ranked. Thus, eliminate the last column, "Total," and the subcolumns Average and Rank. Adjust the main text and the legend accordingly.
Reply: We agree with your comment. The table was modified as suggested, and the text was edited. It now reads: "The treatments' results were analyzed to facilitate their interpretation by comparing their performance in percentages to the control (healthy, non-infected, Captan chemically coated seeds plans, Tables 6 and 7). According to this comparison, the highest performing CRD management practice was chemical seed coating and Azoxystrobin sprinkling (Sp), with sowing, and the least impactful treatment was chemically coated seed plans combined with Azoxystrobin sprinkling followed by irrigation at low dosage along the growth (Sp + D200)" (lines 467-473). We also corrected the Tabels' footnotes.
- Figure 4. Change the title of the Figure, according to what it shows. Shorten the legend. Explain the treatment and the symbols, without recalling other Figures. Each figure must be self-explanatory. Avoid repeating methods details. Add the statistical test used for comparisons. See the comments already given for the other Figures.
Reply: The figure title was corrected to "Real-time PCR analysis of the amount of M. phaseolina DNA in plant roots in the pot experiment on days 29 and 52." The figure's legend was corrected and shortened, as advised.
- Table 8. delete completely. At a maximum, show the results of PCR only, for each sampling day without average or ranking.
Reply: Following your suggestion, the table was edited and now shows only the results of the qPCR comparative analysis.
- Adjust the main text and the legend accordingly.
Reply: Adjusted.
- Figure 5. Shorten the caption, as already pointed out for Figure 1. Avoid methods description etc.
Reply: The figure caption was shortened and focused based on your advice.
- Figure 6 The statistical analysis results should be reported. Add the letters to indicate statistical significance. Adjust the Figure caption as already pointed out for the other Figures.
Reply: Indeed. We repeated the statistical analysis and added the letters to indicate statistical significance to the figure (now Figure 7). Statistically significant differences could only identified in Figure 7B (symptoms evaluation). No statistical differences could be reached in the yield assessment due to the low number of repeats (some field margin plots had irregular growth due to uneven water dispersal, weeds, wind, and other reasons).
- Line 583 dispersal is not the right term.
Reply: The term "dispersal" was replaced by "spreading".
- Line 607-609 How can you say that biocoating gave an advantage in yield and symptoms? Is it statistically supported? I do not see any statistical lettering in figure 6. Add letters and adjust the comments. Avoid to base this statement only on ranking (generally avoid ranking). Based on the Figure 6, yield appears stable, neither affected by the chemical or biological coating, nor by the treatment!
Reply: You are correct; this should be explained better. The paragraph was rewritten and now reads: "The yield assessment didn't allow the identification of statistical differences due to the low number of repeats (some field margin plots had irregular growth due to uneven water dispersal, weeds, wind, and other reasons). Still, most biological treatments outperformed the chemical intervention, enhancing the cotton plants' yield by up to 17% (Figure 7A). These treatments protected against charcoal rot and improved the plants' health under field conditions (up to 31% in the bio-coating and chemical sprinkling treatment, p < 0.05, Figure 7B)." (lines 539-546).
- Symptoms: I am not sure that the biological approach is better than the chemical one, looking at the graph. Add statistics.
Reply: We repeated the statistical analysis and added the information to the graph (Figure 7B). Statistical difference was found in one treatment (compared to two chemical treatments), in the bio-coating and chemical sprinkling treatment (p < 0.05).
- Line 612 interphase is not the right term.
Reply: The term "interphase" was replaced by "control method."
- Figure 7 Rephrase the title and the caption, avoid referring to other Figures, explain the name of the treatments, avoid method description etc.
Reply: The title was corrected to "Real-time PCR analysis of the amount of M. phaseolina DNA in plant roots in the commercial field trial," and the caption text was substructed and focused.
- Table 9 Again, restructure this table, as already suggested, without total, without averaging and without ranking.
Reply: The table was modified as suggested, and the text was edited accordingly. We also corrected the Tabels' footnotes.
- Line 681 milestone appears as an excessive term.
Reply: We agree and replaced this term with "step."
Round 2
Reviewer 3 Report
The manuscript has been improved according to the main issues raised in my first report, but it still needs further revision before being published.
I regret to notice that the authors did not accept all my recommendations. For instance, the Figure captions are still not self-explanatory, refer often to other Figures or Tables, the acronyms are not explained, details of materials and methods are still present etc…
I appreciated very much that the authors inserted letters on the bar charts, making clearer the statistical significance of the results, which was very confusing in the first draft.
As a consequence, the whole picture is clearer now, i. e. the results now are clear, and I have to say that I could not find correspondence between the results (accompanied by statistics) reported in the Figures, and the authors’ statements in the Results section. It appears that the authors completely ignored the results of the statistical analysis during the presentation of the trial results. In general, the authors based their statements on the ranking of the treatments, which were based on the absolute values of the means reported in the Tables.
Thus, I regret to say that the authors must rewrite the presentation of the results, based on the statistics, and change the discussion and conclusions accordingly.
Performing field trials under natural conditions of infection indeed is a challenge, due to the nonhomogeneous presence of the pathogens, and the disease, thus it is difficult to obtain statistically significant results. However, the repetition of the field trial in different years and a correct statistical analysis over the years can help in reaching statistically sound results, which allow to drive scientific conclusions. In the case of experiments with artificial infections under controlled conditions, this goal can be achieved more easily, but it is recommended to repeat the trial to confirm the results. In this work, both trials were not repeated, and the treatments very often were not statistically different in both cases, thus the authors must be more cautious and avoid indicating which was the best or the worst treatment.
Anyway, as I already said, I always appreciate studies that are not only limited to laboratory conditions but go further up to the field level.
Thus, I would like to give my further suggestions to help the authors in making the manuscript at least acceptable from a scientific point of view, whatever the results obtained.
My suggestions:
Abstract and introduction
Line 17 and 129 add agents after biological, as done in the title
Line 32 substitute “to the combined” with “about the combined”
Line 137 check the Trichoderma code
Line 148 and 166 As in the previous report, I do not understand the use of the term interphase. Please delete interphase here and everywhere in the text
Line 149 substitute interphase with method
Line 151 delete demonstrating and substitute it with “which demonstrated”
Line 155 delete “oppress and”
Line 135,157check the italics for Trichoderma, here and throughout the manuscript
Line 159 after growth stage, add “in pots, under controlled conditions, and in a commercial field”.
Delete the last lines from 163 up to 166
Line 168 Fungal sources (plural, because there are the pathogen and the Trichoderma strains)
Line 171, add where this Research Centre is located in Israel
Line 189 Rename the title into Pot assay in growth room
Line 191 the study was conducted in pots in a growth room
Line 192-197 delete all and substitute with: “The experiment included 2 types of seed coating (biological or chemical) as controls, and 5 different applications of the fungicide Azoxystrobin (Amistar S.C.; Syngenta, Basel, Switzerland, supplied by Adama Makhteshim, Ashdod, Israel), in combination with each type of seed coating, for a total of 12 treatments. Such treatments are listed in Table 2, along with the dosage of fungicide and time of application”.
To complete the treatment presentation, describe here also the biological and chemical treatment method used for this assay in pots. “The biological coating was performed….The chemical coating was performed….
After the number of replications, give here also the experimental design used to collocate the pots in the growth room (fully randomized design, as cited in the statistical analysis chapter)
Line 198 An infected control is missing. It would have been useful to evaluate the efficacy of the artificial infection with the pathogen and the efficacy of the biological or chemical coating in protecting the plants from the infection, comparing these coating treatments with the infected control.
Line 201 Change the caption of table 2 into “Trial architecture of the pot assay in growth room: description of the different treatments, fungicide dosage and application time”
Table 2 Substitute Application with Treatment in the column titles
Delete “with Captan” in the first row, as in the note it is reported that Captan was not the only chemical used. Leave just Chemical seed coating. The note explains the fungicides used. Add a note for the biological coating
Figure 1 IS TO BE DELETED The figure is a pictorial description of the experimental scheme already reported in Table 1. Unfortunately, it confounds the reader because it is not clear that the seed coating is always done (alone or in combination). In addition, the figure needs a long legend to understand the design. I have already recommended that the authors not repeat materials and methods details in the Legends of Figures and not refer to other tables or figures. Since Table 1 clearly and fully describes the experimental design, I recommend the authors delete this figure, because it occupies space and does not add new information.
Line 220 I am not sure that according to [xx] is an accepted notation
Line 220-224 Delete. It is a discussion, not a material and method’s description. Start with Pots…
Line 227 substitute “ground” with “soil”. Gound has a different meaning
Line 246-256 TO BE DELETED. This treatment description is very very confusing. There is also a sort of discussion about the double dosage of fungicide and citation of the field assay that has not been yet presented. The treatments have been already introduced in the trial architecture and Table 2. It is useless to repeat here. Just save the note on the mock group and integrate into the description of the trial architecture.
As a consequence, change the title of the 2.2.3 paragraph into Macrophomina phaseolina infection
Line 257 Here start a new paragraph that can be 2.2.4 Experimental determinations, and describe all the determinations performed with methodology details (not only the list): besides emergence percentage, assessment of symptoms (how?), growth parameters (which? they are presented only in the Results), survival percentage or rate (NOT index), and pathogen DNA content in the roots (how?)
(By the way, I did not find any results about symptoms in the Results of the pot assay).
Line 267 substitute “estimated” with “studied” (also in other paragraphs), and add information on the “common” chemical approach. Which chemicals are commonly used? Or do you refer to the chemical approach used here? It is not clear
Line 278 and following: substitute meters with m
Line 282 delete (Captan)
Line 310-331 Some details can be moved to the paragraph that describes the biological coating in the pot assay. Here the authors could describe the scaling process up to the field level (volumes will be higher)
Line 374-375 Correct into: using aerial imaging instruments onboard unoccupied aerial vehicles (UAVs).
Line 388-389 delete and move in the corresponding trial architecture paragraph
Line 400 Pot assay in growth room
Line 402-403 Delete “for the first time in Israel in a commercial field ”
Figure 2 legend: Substitute like this: Pot experiment. Thinning from five to one plant per pot on day 29 post-sowing (A). A representative well-developed plant on day 29 post-sowing (D). Plants at harvest on day 52 post-sowing (E, F).
Line 413 Looking at the growth parameters measured at midseason (Figure 2 A-D)
Line 414-415 the authors write that chemical coating with Azoxystrobin sprinkling at seeding (Sp) provided the highest control over charcoal rot (Figure 3). This is false. This figure shows growth parameters, not disease symptoms, thus how can the authors say that one treatment provided control against the disease? They should comment on this figure in terms of growth stimulation or inhibition, not disease control. Moreover, the statistical analysis did not highlight any significant positive effect of any of the treatments on the growth parameters. Please present the results based on the statistical analysis. Sp treatment did not “outperform” the other treatments for root weight. It just provided the highest absolute mean value, which was not statistically different from 6 other treatments.
Instead, negative effects are highlighted on shoot weight and height, number of leaves due to Sp+D200 and sp+D400 treatments.
Line 415-417 Again, there is no improvement in root weight or survival, since these values/percentages are not statistically different. Delete all these considerations.
Line 419-430 Again, the biological control-based managements were NOT less effective, since no significant differences were highlighted. Biologically treated seeds without M. phaseolina infection did NOT yielded lower growth parameters than chemically treated ones (as above). Etc…
Line 415 delete sprouts’ . It is root fresh weight or dry weight? Specify
Line 431-432 This is supported by statistics, thus cite Figure 3
Line 433-434 This is not true for all the parameters cited. Check the letters in Figure 3.
Figure 3
Line 438-440 Correct like this: Growth parameters and survival rate of the plants after the different treatments in the pot assay on day 29. Shoot fresh weight (A); shoot height (B); etc…and survival rate (F).
Line 441 delete “The experiment is described in Figure 1”. I had already recommended the authors not to refer to other Figures or Tables in the legends
Line 443 delete (Table 2)
Line 446 add (Non-infe) after ”chemical coating”. Delete “The experiment included etc…per treatment)”
Line 448 delete assessment
Line 447 the standard error
Line 449 correct signified into shown
Line 453-459 There are no significant differences among the treatments thus present the results accordingly, avoiding presenting the best and the second classified
Line 461, Correct the legend as suggested for Figure 3. Repeat the acronym explanation exactly as done for figure 3. Delete “All the acronyms and the treatments' names are as in Figure 3”. The figure must be self-explanatory.
Line 467-472 These lines must be eliminated. The results obtained for the growth parameters of the pot assay have been correctly analysed by ANOVA and presented by the bar chart. Table 6 reports the same data as Figures 3 and 4, expressed in percentage of the healthy noninfected control (without statistics). I suggest moving the table as Supplementary material, so that the readers can get the data more easily than from the bar chart. But these data do not add further information than Figure 3 and 4. This table can be cited in brackets when Figure 3 and 4 are cited (es: Figure 3, Table SX).
Line 474 Please change into Table 6. Growth parameters and survival rate of the plants after the different treatments in the pot assay on day 29 and 52 post-sowing, expressed as percentage of healthy, noninfected controls
Add a note explaining the treatment acronyms, as done for Figure 3 and 4. Do not add any comment
Line 475-479 To be deleted
Line 480-482 Correct into The variation of M. phaseolina DNA in the root tissue of the plants sampled at mid-season (day 29) and at the end of the experiment (day 52) after the different treatments is reported in Figure 5.
Line 483-490 delete all this, because it is not statistically supported. The presentation of the results must be adherent to the statistics shown in the bar charts. I can see that the treatment Sp+D400 increased the pathogen infection in combination with chemical coating at 29 day, and in combination with biological coating at 52 day (the only 2 treatments that differed statistically from the others[a]).
Line 493 Figure 5 Variation of Macrophomina phaseolina DNA in the root tissue of plants sampled on day 29 (A) and 52 (B) post-sowing after the different treatments in the pot experiment.
Line 494-495 is to be deleted. Describe the acronyms as for the other Figures
Line 497-499 This information must be given in the text, not here in the legend.
Line 501 shown, instead of signified
Table 7 Move to the Supplementary materials, for the same reasons as for Table 6. Correct the title and the note as suggested for Table 6
Line 517 up to age 74 correct into up to 74 days post-sowing
Line 518-519 delete as expected [30]. It is not the discussion chapter
Line 528-535 Figure 6. Correct into Commercial field trial experiment. Plant at mid-season, 74 day post sowing (A-C). Longitudinal section of a healthy (D, upper) and Macrophomina phaseolina infected (D, lower) plant stem. Symptoms of plant wilting (E). Cross section of a diseased stem (F).
Line 522-524. Correct into The typical CRD symptoms involving vascular bundle browning, resulting from toxins interacting with a plant-response product, likely gossypol, were observed (Figure 6 D-F)
Line 524-526 delete. It is not a result
Line 537-539. This is a summary of the results that should be moved after having presented the results for yield symptoms and pathogen DNA content in the roots, (once evidenced the statistically significant superior of this treatment).
Line 540 the number of plots (8 replicates) reported in the materials and methods section appeared adequate. It should not be the cause of the absence of statistical significance.
Line 542-543 For yield, the “outperforming” of the biological treatments (up to 17%) cannot be visually caught in the chart, being not statistically significant. You can say that up to a 17% increase in yield was observed for biological treatments, even if not statistically significant.
Line 543-545 For Symptoms, looking at the statistics, you can only say that the Biological coating+Sp lowered symptoms compared to Chemical coating+D200 and Chemical coating +Sp+D200, but this treatment (Bio+Sp) was not different from all the other treatments, biological or chemical. Words are important.
546-547 Near harvest, Biocoating+Sp+D400 reduced the pathogen DNA compared to Chemical coating+Sp+D400, but it was not different from all the other treatments (looking at statistics)
Line 559 The symptoms evaluation must be described in the Materials and Method section. There it was reported that the disease impact was assessed by counting the number of dead plants. Here in the Figure 7, the symptoms are ranked on a scale 0-20, instead. Please correct or add the necessary information in the MM section.
Figure 7 and 8 The title and legend must be corrected according to the already given recommendations.
Line 550-552 Please explain what the authors intend for traditional chemical intervention. Which data were used to make this t-test? Where are these data? It is not clear.
Line 552-554 Where are the percentages of diseased plants? And the number of wilted plants?
Table 8. Move this table among Supplementary materials, correct Title and note as above recommended
Line 587-588 use the acronyms and cite the Figure, to look on the left and on the right
Line 590 -594 This is not correct. Table 8 reports the mean values calculated over all the plots that received the same treatment, I guess, thus the values of 89% and 69% are mean and not the individual values of the single plots 44 and 53. If the authors want to compare these two plots, they should compare the individual values of disease degree and aerial index of these plots.
Figure 9 Adjust the title: Aerial imaging of the commercial field trial at 162 days post-sowing, obtained by aerial remote sensing: high-resolution visible range (RGB, A), green–red vegetation index (GRVI, B), and thermal imaging (C). Two adjacent representative plots in the middle of the field are shown: plot 44 on the left treated with Trichoderma spp. mix seed coating + Azoxystrobin sprinkling (Sp), and plot 53 on the right, treated with Trichoderma spp. mix seed coating + Azoxystrobin irrigation at low dose treatment (D200).
Comments to the figure should be mentioned in the text, not in the legend
Table 9 Correct the title and the note of the table as suggested for the other tables. These data are not duplicated by a bar chart, like in the cases above, thus the table can remain here. What about data related to RGB? They are missing in the table.
The discussion is too long and not very focused on the results obtained. Some paragraphs can be deleted, because are too general. Try to reduce the length. See also below.
Line 616 correct condition into disease
Line 618-621 Delete
Line 632-642 please summarize
Line 653 Taking together, the current etc..
Line 649-650 correct into: it lessens the impact of chemical treatments on health and environment
Line 655-656 It is not supported by the statistic analysis. It was not evident.
Conclusion
Line 762-763 Correct into: Both trials aimed at evaluating the effectiveness of an integrated biological/chemical approach, i.e. based on biological seed coating with Trichoderma spp. followed by the application of the fungicide Azoxystrobin, in comparison to only biological seed coating, or a completely chemical approach, based on only seed coating or in combination with Azoxystrobin.
Line 764- 767 These statements are not statistically supported, thus I recommend the authors to drive conclusions based on the statistical significance. They must rewrite the sentences, according to the statistical analysis results.
In general, the authors must seriously reflect on the real results of this work and not be driven by what they would have liked to obtain, that is the hypothetical superiority of the integrated approach (biological+chemical). This superiority does not come out from the results of this work, based on one trial in pots and one in field. Maybe they need to repeat the trials.
It could be instead interesting to compare the performance of the biological or integrated approach vs the completely chemical.
In pots, at 29 days, comparing biological coating vs chemical coating, either alone or within each treatment (i.e., the green vs the yellow bars) I can see that the green bars were always NOT different from the yellow (i.e. biocoating not different from chemical coating and bio-chemical integration not different from chemical) for all the growth parameters. Unique exceptions: shoot height and number of leaves are stimulated by biocoating+Sp+D200 compared to the corresponding chemical coating+Sp+D200.
At 52 days, it is the same, but the unique exception is valid only for shoot height.
For root infection, at 29 days, no differences. At 52 days, chemical coating+D400 is more protective than the corresponding biocoating+D400. The same occurs for Chemical+Sp+400 which is superior to the corresponding biocoating+Sp+D400.
In the field, no differences at all among the treatments for yield, and no differences among bio+chem and chem for symptoms at 151 days. This is interesting because it means that the seed coating only with Trichoderma provides enough protection to sustain yield and control CRD, as well as all the other treatments differently combined! A great result from an economic and environmental point of view.
About the root infection, no differences between biocoating and chem coating or bio+chem and completely chem approach at 74 days. At 151 days it is the same, except for biocoat+Sp+400 which was superior to the corresponding chem+Sp+D400
Thus, I will finally suggest shifting the work in this direction, instead of trying to find the best option among the treatments, because the results do not allow to find the best option.

Author Response
Responses to Reviewer 3's comments
We thank the reviewer for investing substantial effort, which undoubtedly contributed to this manuscript. The remarks and suggestions improved this paper's scientific soundness and accuracy. Your contribution is greatly appreciated.
Major comments
Some methods are missing. The experiment was not repeated.
Reply: It's true, but conducting a full-season experiment in a cotton field can take nearly six months and is only feasible once a year. This process demands significant investment in labor and other expenses but yields a substantial dataset on plant growth, yield parameters, molecular analysis, and remote sensing measurements collected from the air. The current fieldwork essentially replicates the 2022 study (referenced here: https://doi.org/10.3389/fpls.2023.1272335) in the same field, with separate testing of biological and chemical treatments. These two studies complement each other and introduce valuable new insights. To address the lack of repetition in the field experiment and to maintain a manageable number of results for a single article, we conducted a parallel experiment in a controlled growing room. This indoor experiment closely mimicked the conditions of the field study design. We believe that combining the results of both experiments conducted here provides crucial information for establishing and expanding upon our findings in future research.
The results have been statistically analyzed, but many statements are not supported by the statistics shown.
Reply: We carefully followed the specific remarks below and edited the entire Result section so that the presentation of the results will focus on those that gained statistical significance.
The manuscript has been improved according to the main issues raised in my first report, but it still needs further revision before being published.
Reply: Thank you for the positive evaluation of our manuscript. As detailed below, all the remarks and suggestions were addressed carefully and thoroughly.
I regret to notice that the authors did not accept all my recommendations. For instance, the Figure captions are still not self-explanatory, refer often to other Figures or Tables, the acronyms are not explained, details of materials and methods are still present etc…
Reply: All the reviewer recommendations were addressed carefully, and we revised the manuscript to address the specific comments and suggestions fully, as detailed in our point-by-point response below.
I appreciated very much that the authors inserted letters on the bar charts, making clearer the statistical significance of the results, which was very confusing in the first draft. As a consequence, the whole picture is clearer now, i. e. the results now are clear.
Reply: Thank you.
I have to say that I could not find correspondence between the results (accompanied by statistics) reported in the Figures, and the authors' statements in the Results section. It appears that the authors completely ignored the results of the statistical analysis during the presentation of the trial results. In general, the authors based their statements on the ranking of the treatments, which were based on the absolute values of the means reported in the Tables. Thus, I regret to say that the authors must rewrite the presentation of the results, based on the statistics, and change the discussion and conclusions accordingly.
Reply: You're right. We corrected the text and rewritten the presentation of the results, as detailed below.
Performing field trials under natural conditions of infection indeed is a challenge, due to the nonhomogeneous presence of the pathogens, and the disease, thus it is difficult to obtain statistically significant results. However, the repetition of the field trial in different years and a correct statistical analysis over the years can help in reaching statistically sound results, which allow to drive scientific conclusions. In the case of experiments with artificial infections under controlled conditions, this goal can be achieved more easily, but it is recommended to repeat the trial to confirm the results. In this work, both trials were not repeated, and the treatments very often were not statistically different in both cases, thus the authors must be more cautious and avoid indicating which was the best or the worst treatment. Anyway, as I already said, I always appreciate studies that are not only limited to laboratory conditions but go further up to the field level.
Reply: We agree. We did our best to be more cautious in interpreting the results. See our detailed point-by-point response below.
Thus, I would like to give my further suggestions to help the authors in making the manuscript at least acceptable from a scientific point of view, whatever the results obtained.
Reply: We very much appreciate your assistance in enhancing the article. We deeply value the thorough and constructive advice provided, which will undoubtedly aid us in crafting future articles. Thank you.
Specific Comments:
Line 17 and 129 add agents after biological, as done in the title.
Reply: We added "agents" as advised.
Line 32 substitute "to the combined" with "about the combined".
Reply: Corrected as advised.
Line 137 check the Trichoderma code.
Reply: The Trichoderma strain number correct to T7507.
Line 148 and 166 As in the previous report, I do not understand the use of the term interphase. Please delete interphase here and everywhere in the text.
Reply: The term " interphase" deleted from the text (in 4 places).
Line 149 substitute interphase with method
Reply: Done.
Line 151 delete demonstrating and substitute it with "which demonstrated."
Reply: Corrected.
Line 155 delete "oppress and."
Reply: Deleted.
Line 135,157check the italics for Trichoderma, here and throughout the manuscript.
Reply: Checked and corrected throughout the text.
Line 159 after growth stage, add "in pots, under controlled conditions, and in a commercial field".
Reply: Added.
Delete the last lines from 163 up to 166.
Reply: Deleted.
Line 168 Fungal sources (plural, because there are the pathogen and the Trichoderma strains).
Reply: Indeed. Corrected.
Line 171, add where this Research Centre is located in Israel.
Reply: Added.
Line 189 Rename the title into Pot assay in growth room.
Reply: The title corrected.
Line 191 the study was conducted in pots in a growth room.
Reply: We added " in pots" to the description as advised.
Line 192-197 delete all and substitute with: "The experiment included 2 types of seed coating (biological or chemical) as controls, and 5 different applications of the fungicide Azoxystrobin (Amistar S.C.; Syngenta, Basel, Switzerland, supplied by Adama Makhteshim, Ashdod, Israel), in combination with each type of seed coating, for a total of 12 treatments. Such treatments are listed in Table 2, along with the dosage of fungicide and time of application".
Reply: The text was corrected according to your advice. Thank you.
To complete the treatment presentation, describe here also the biological and chemical treatment method used for this assay in pots. "The biological coating was performed…. The chemical coating was performed….
Reply: Section 2.3.3. (The biological protective treatment) was moved and integrated with the pot assay description. The following explanation was added to the text: "Bio-shielding was carried out in seed coating, as described in [9]. The same bio-coated seeds batch was used for the growth room trial and the field assay (see section 2.2.2 for the detailed preparation protocol for the seeds coating). The seed company (Israel Seeds) performed the chemical coating as a customary treatment for all non-organic seeds used for commercial crop production." (lines 204-209).
After the number of replications, give here also the experimental design used to collocate the pots in the growth room (fully randomized design, as cited in the statistical analysis chapter).
Reply: As suggested, we added the experimental design: "The treatments and control pots were arranged in the growth room in a fully randomized design" (lines 211-212).
Line 198 An infected control is missing. It would have been useful to evaluate the efficacy of the artificial infection with the pathogen and the efficacy of the biological or chemical coating in protecting the plants from the infection, comparing these coating treatments with the infected control.
Reply: You are right; such an addition may provide benefit information. Nevertheless, there is an infected control, the chemically coated seeds. These seeds undergo standard general pesticide treatment to reduce various pathogens' pressure during the plants' first developmental stage. Yet, the common pesticide treatment with Captan had no adequate protection against CRD (see reference to the subject in lines 200-202). Since the field experiment was conducted with such chemically treated seeds (according to the requirement of the farmers since it is not an organic cultivated field), the pot's trial was conducted in the same way.
Line 201 Change the caption of table 2 into "Trial architecture of the pot assay in growth room: description of the different treatments, fungicide dosage and application time".
Reply: The title was replaced as suggested.
Table 2 Substitute Application with Treatment in the column titles
Reply: Substituted.
Delete "with Captan" in the first row, as in the note it is reported that Captan was not the only chemical used. Leave just Chemical seed coating. The note explains the fungicides used. Add a note for the biological coating
Reply: The first row (Table 2) was corrected as advise. The Biological coating description was added to the table foot notes: "Fungal spores, mycelium fragments, and extrolites were harvested from fungal colonies and mixed with Tween 80 (0.05%). This coating solution was mixed with cotton seeds (not chemically treated) for ten min."
Figure 1 IS TO BE DELETED The figure is a pictorial description of the experimental scheme already reported in Table 1. Unfortunately, it confounds the reader because it is not clear that the seed coating is always done (alone or in combination). In addition, the figure needs a long legend to understand the design. I have already recommended that the authors not repeat materials and methods details in the Legends of Figures and not refer to other tables or figures. Since Table 1 clearly and fully describes the experimental design, I recommend the authors delete this figure, because it occupies space and does not add new information.
Reply: This Figure was specifically requested by reviewer 1. It is indeed a pictorial description of Table 2, but reviewer 1 wrote: "Additionally, it is recommended to include illustrative diagrams of the treatments for better clarity. " We suggest leaving the decision rather than keeping it, moving it to the supplementary material, or deleting it to the Editor. Regarding your concern about the unclarity of the seed coating treatments, we improved the illustration; hopefully, it is better self-explanatory now.
Line 220 I am not sure that according to [xx] is an accepted notation
Reply: This sentence was omitted from the text. We added the reference details in another place when the phrase was used: "according to Jans et al., 2021 [2]) " (line 42).
Line 220-224 Delete. It is a discussion, not a material and method's description. Start with Pots…
Reply: Deleted as suggested.
Line 227 substitute "ground" with "soil". Gound has a different meaning.
Reply: Substituted.
Line 246-256 TO BE DELETED. This treatment description is very very confusing. There is also a sort of discussion about the double dosage of fungicide and citation of the field assay that has not been yet presented. The treatments have been already introduced in the trial architecture and Table 2. It is useless to repeat here. Just save the note on the mock group and integrate into the description of the trial architecture.
Reply: The paragraph was deleted and the note on the mock group was integrated into the description of the trial architecture.
As a consequence, change the title of the 2.2.3 paragraph into Macrophomina phaseolina infection.
Reply: The 2.2.3 paragraph title was corrected accordingly.
Line 257 Here start a new paragraph that can be 2.2.4 Experimental determinations, and describe all the determinations performed with methodology details (not only the list): besides emergence percentage, assessment of symptoms (how?), growth parameters (which? they are presented only in the Results), survival percentage or rate (NOT index), and pathogen DNA content in the roots (how?).
Reply: We edited the text according to your remarks. The paragraph now reads: "The above-ground emergence percentages in each treatment were checked after ten days. Survival percentages (counting the living sprouts), growth indices (roots and shoot weight, plants above ground height, and the leaves number), and fungus DNA content in each plant's root using qPCR were determined on days 29 and 52 past sowing (detailed in 2.4)." (lines 285-289).
(By the way, I did not find any results about symptoms in the Results of the pot assay).
Reply: You are right; there were no clear visible wilting symptoms during the sprouting phase. This is expected since the disease is latent at this phase and outbursts later from flowering onwards. Still, careful examination of the survival and plants' development parameters (weight, height, and leaf count) reveals the pathogen influence. Even more so, the qPCR analysis shows the pathogen establishment within the plants.
Line 267 substitute "estimated" with "studied" (also in other paragraphs), and add information on the "common" chemical approach. Which chemicals are commonly used? Or do you refer to the chemical approach used here? It is not clear.
Reply: The term "estimated" was replaced with "studied" here and in two other places. The following explanation was added to the text: " The standard chemical intervention relies on general pest control using Captan seed coating and, in severe CRD-infected fields, adding Azoxystrobin or other fungicides to the sowing strip or during the season through irrigation." (lines 300-303).
Line 278 and following: substitute meters with m.
Reply: Corrected here and in a few other places.
Line 282 delete (Captan).
Reply: Deleted.
Line 310-331 Some details can be moved to the paragraph that describes the biological coating in the pot assay. Here the authors could describe the scaling process up to the field level (volumes will be higher).
Reply: Section 2.3.3. (The biological protective treatment) was moved and integrated with the pot assay description. The following explanation was added to the commercial field trial architecture explanation: "The same bio-coated seeds batch was used for the growth room trial and the field assay (see section 2.2.2 for the detailed preparation protocol for the seeds coating)." (lines 205-207).
Line 374-375 Correct into: using aerial imaging instruments onboard unoccupied aerial vehicles (UAVs).
Reply: Corrected as advised.
Line 388-389 delete and move in the corresponding trial architecture paragraph.
Reply: Done as suggested.
Line 400 Pot assay in growth room.
Reply: The title (section 3.1) was corrected as advised.
Line 402-403 Delete "for the first time in Israel in a commercial field"
Reply: Deleted.
Figure 2 legend: Substitute like this: Pot experiment. Thinning from five to one plant per pot on day 29 post-sowing (A). A representative well-developed plant on day 29 post-sowing (D). Plants at harvest on day 52 post-sowing (E, F).
Reply: Corrected. Thank you.
Line 413 Looking at the growth parameters measured at mid-season (Figure 2 A-D)
Reply: The sentence was replaced according to your suggestion.
Line 414-415 the authors write that chemical coating with Azoxystrobin sprinkling at seeding (Sp) provided the highest control over charcoal rot (Figure 3). This is false. This figure shows growth parameters, not disease symptoms, thus how can the authors say that one treatment provided control against the disease? They should comment on this figure in terms of growth stimulation or inhibition, not disease control.
Reply: You are right; referring to growth stimulation or inhibition is indeed more accurate. While this has been said, it should be remembered that during the sprouting growth stages (up to the flowering), the disease is expressed almost exclusively in growth reduction (actual wilting symptoms are absent). Thus, growth suppression (and even more so seedlings death, the survival rate) is, in fact, a disease symptom. We edited the text to express this better: "Looking at the growth parameters measured at midseason (Figure 2, A-C), the chemical coating with Azoxystrobin sprinkling at seeding (Sp) implies some control over charcoal rot-related growth inhibition in terms of absolute mean value without a statistically meaningful difference (Figure 3, Table 6)." (lines 428-432).
Moreover, the statistical analysis did not highlight any significant positive effect of any of the treatments on the growth parameters. Please present the results based on the statistical analysis. Sp treatment did not "outperform" the other treatments for root weight. It just provided the highest absolute mean value, which was not statistically different from 6 other treatments. Instead, negative effects are highlighted on shoot weight and height, number of leaves due to Sp+D200 and sp+D400 treatments.
Reply: We agree and corrected the entire paragraph accordingly: "Because the disease often remains latent in sprouts, it is difficult to achieve statistically significant differences in the growth parameters of the plants, as was the case here. Looking at the growth parameters measured at midseason (Figure 2, A-C), the chemical coating with Azoxystrobin sprinkling at seeding (Sp) implies some control over charcoal rot-related growth inhibition in terms of absolute mean value without a statistically meaningful difference (Figure 3, Table 6). This treatment improved the sprouts' root weight by 21% compared to the non-infected control (Figure 3C). In contrast, negative effects are highlighted on shoot weight and height, number of leaves due to Sp+D200 and sp+D400 treatments combined with chemical seed coating (p < 0.05)." (lines 427-435).
Line 415-417 Again, there is no improvement in root weight or survival, since these values/percentages are not statistically different. Delete all these considerations.
Reply: The sentences were deleted as suggested, leaving only the most notable differences caused by the treatments.
Line 419-430 Again, the biological control-based managements were NOT less effective, since no significant differences were highlighted. Biologically treated seeds without M. phaseolina infection did NOT yielded lower growth parameters than chemically treated ones (as above). Etc…
Reply: We agree. The entire paragraph was corrected, and it now focuses on those treatments that gain statistically significant differences from the control (lines 436-452):
"The biological control-based integrated management had a similar impact to the above treatments. These treatments had a lower starting point since biologically treated seeds without M. phaseolina infection yielded lower growth parameters than chemically treated ones, used here as a control and a reference (Tables 6 and 7). A significant improvement over the bio-coting healthy control can be seen in the Trichoderma species' mixture seed coating alone (SC only) shoot weight (44%, p < 0.005, Figure 3A). In the shoot weight assessment, this treatment (biological SC only) outperformed (p < 0.05) all biological-based treatments and all Azoxystrobin-based irrigation treatments. The biological SC Also achieved statistically discernible improvement in the root weight and leaves count values, compared to the biological mock control, the D400 and the Sp+D200 and sp+D400 treatments (Figure 3 C-E).
Evidentially, all chemical treatments that included both Azoxystrobin sprinkling and irrigation (Sp + D200 and Sp + D400) led to 15-29% growth suppression (Table 7), with significant reduction (p < 0.05, Figure 3 A- B, E) in the shoot development when combined with chemo seed coating most likely due to a phytotoxicity resulting from overdose. Interestingly, in the biological seed-coating plants, the double Azoxystrobin treatment (Sp + D200) was significantly less harmful (p << 0.05)."
Line 415 delete sprouts'. It is root fresh weight or dry weight? Specify
Reply: The entire paragraph was rewritten (see our reply above).
Line 431-432 This is supported by statistics, thus cite Figure 3
Reply: The paragraph was corrected according to your remark: "Evidentially, all chemical treatments that included both Azoxystrobin sprinkling and irrigation (Sp + D200 and Sp + D400) led to 15-29% growth suppression (Table 7), with significant reduction (p < 0.05, Figure 3 A- B, E) in the shoot development when combined with chemo seed coating most likely due to a phytotoxicity resulting from overdose." (lines 447-450).
Line 433-434 This is not true for all the parameters cited. Check the letters in Figure 3.
Reply: See our reply to your remark above. We edited the paragraph and specified that statistically significant shoot development reduction was documented when Azoxystrobin sprinkling and irrigation (Sp + D200 and Sp + D400) treatments were combined with chemo seed coating.
Line 438-440 Correct like this: Growth parameters and survival rate of the plants after the different treatments in the pot assay on day 29. Shoot fresh weight (A); shoot height (B); etc…and survival rate (F).
Reply: Corrected as advised.
Line 441 delete "The experiment is described in Figure 1". I had already recommended the authors not to refer to other Figures or Tables in the legends.
Reply: We deleted the sentence "The experiment is described in Figure 1."
Line 443 delete (Table 2)
Reply: Deleted.
Line 446 add (Non-infe) after" chemical coating". Delete "The experiment included etc…per treatment)"
Reply: We added (Non-infe) after the control wish is a more precise description: "The treatments and the control (Non-infe) were performed once with chemical seed coating (Chemical SC) and once with bio-coating (Trichoderma species mix, Biological SC)." (lines 456-458).
Since the number of repeats may change between the figures, we specified it in each case. We added an explanation in the Materials and methods: "The specific repeat number used to analyze the results is indicated in each figure. " (line 210).
Line 448 delete assessment.
Reply: Deleted.
Line 447 the standard error.
Reply: Corrected.
Line 449 correct signified into shown.
Reply: Corrected.
Line 453-459 There are no significant differences among the treatments thus present the results accordingly, avoiding presenting the best and the second classified
Reply: The paragraph was edited as suggested, and it now reads: "At growth day 52 (the experiment end, Figure 2 E, F), the growth indices were generally similar to those recorded on day 29 but had less statistical significance (Figure 4, Table 6). Here, the chemical coating (Captan) with Azoxystrobin sprinkling at seeding (Sp) marked the highest increase in root and shoot biomass (38% and 45%, Figure 4 A, C) and leaves count (18%, Figure 4E). Bio-coating (as a sole treatment) improved root and shoot weight (37% and 34%, respectively). The biological-based integrated treatment (bio-coating + Azoxystrobin sprinkling, D200) was most impactable regarding the root length (20% improvement over the control). As in sampling day 29, a statistically significant reduction (p < 0.05) was measured in the double Azoxystrobin treatment (Sp + D200, Sp + D400) combined with chemical seed coating regarding the shoot weight (Figure 4A). The Sp + D200 (with the chemical seed coating) also led to a significant (p < 0.05) reduction in the shoot height and the leaves number." (lines 466-477).
Line 461, Correct the legend as suggested for Figure 3. Repeat the acronym explanation exactly as done for figure 3. Delete "All the acronyms and the treatments' names are as in Figure 3". The figure must be self-explanatory.
Reply: Agreed. Corrected.
Line 467-472 These lines must be eliminated. The results obtained for the growth parameters of the pot assay have been correctly analysed by ANOVA and presented by the bar chart. Table 6 reports the same data as Figures 3 and 4, expressed in percentage of the healthy noninfected control (without statistics). I suggest moving the table as Supplementary material, so that the readers can get the data more easily than from the bar chart. But these data do not add further information than Figure 3 and 4. This table can be cited in brackets when Figure 3 and 4 are cited (es: Figure 3, Table SX).
Reply: We agree. The text regarding the table was eliminated, and the table was cited along with Figures 3 and 4, as advised. The other reviewers didn't recommend moving the table from the main text, so we prefer to leave Table 6 in its current position (we believe it could help evaluate the impact of the treatments).
Line 474 Please change into Table 6. Growth parameters and survival rate of the plants after the different treatments in the pot assay on day 29 and 52 post-sowing, expressed as percentage of healthy, noninfected controls. Add a note explaining the treatment acronyms, as done for Figure 3 and 4. Do not add any comment.
Reply: Corrected as advised.
Line 475-479 To be deleted
Reply: The original text of Table 6 footnotes was deleted and replaced per your above suggestion.
Line 480-482 Correct into The variation of M. phaseolina DNA in the root tissue of the plants sampled at mid-season (day 29) and at the end of the experiment (day 52) after the different treatments is reported in Figure 5.
Reply: Corrected as advised.
Line 483-490 delete all this, because it is not statistically supported. The presentation of the results must be adherent to the statistics shown in the bar charts. I can see that the treatment Sp+D400 increased the pathogen infection in combination with chemical coating at 29 day, and in combination with biological coating at 52 day (the only 2 treatments that differed statistically from the others[a]).
Reply: We edited the text according to your advice but left some of the information regarding Table 7.
Line 493 Figure 5 Variation of Macrophomina phaseolina DNA in the root tissue of plants sampled on day 29 (A) and 52 (B) post-sowing after the different treatments in the pot experiment.
Reply: The title of Figure 5 was corrected as advised.
Line 494-495 is to be deleted. Describe the acronyms as for the other Figures
Reply: Corrected as advised.
Line 497-499 This information must be given in the text, not here in the legend.
Reply: The information was moved to the text and edited: "On day 52 data, there were 4-5 repetitions in a few treatments due to outlier substruction (identified using the ROUT method in GraphPad software with the Q = 1% recommended setting) [43]. This restriction leads to a lesser statistical analysis power." (lines 498-500).
Line 501 shown, instead of signified.
Reply: Corrected as advised.
Table 7 Move to the Supplementary materials, for the same reasons as for Table 6. Correct the title and the note as suggested for Table 6
Reply: Table 7 was edited per your suggestion, but we prefer to leave it in the main text (see explanation regarding Table 6).
Line 517 up to age 74 correct into up to 74 days post-sowing
Reply: Corrected as advised.
Line 518-519 delete as expected [30]. It is not the discussion chapter
Reply: Deleted.
Line 528-535 Figure 6. Correct into Commercial field trial experiment. Plant at mid-season, 74 day post sowing (A-C). Longitudinal section of a healthy (D, upper) and Macrophomina phaseolina infected (D, lower) plant stem. Symptoms of plant wilting (E). Cross section of a diseased stem (F).
Reply: Corrected as advised.
Line 522-524. Correct into The typical CRD symptoms involving vascular bundle browning, resulting from toxins interacting with a plant-response product, likely gossypol, were observed (Figure 6 D-F)
Reply: Corrected as advised.
Line 524-526 delete. It is not a result
Reply: Agreed. We integrated this explanation in the Introduction (lines 62-64): "One common CRD symptom involves vascular bundles browning resulting from toxins interacting with a plant-response product, likely gossypol [4]. Gossypol, a significant sesquiterpene phytoalexin in cotton, is essential for defending the plant against invading pathogens.
Line 537-539. This is a summary of the results that should be moved after having presented the results for yield symptoms and pathogen DNA content in the roots, (once evidenced the statistically significant superior of this treatment).
Reply: right. The summary was moved as suggested (see lines 568-570): "To summarize, at the season's conclusion, the yield valuation, symptoms assessment, and qPCR-based molecular tracking demonstrate the advantage of the seeds' bio-coating alone or combined with a low dose of Azoxystrobin (Figures 7 and 8, Table 8)."
Line 540 the number of plots (8 replicates) reported in the materials and methods section appeared adequate. It should not be the cause of the absence of statistical significance.
Reply: As explained in Figure 7 legend, each treatment group and the control group were replicated eight times. Still, some field margin plots had irregular growth (due to uneven water dispersal, weeds, wind, and other reasons). Thus, 3-6 plots per treatment were sampled at harvest. This weakens the power of statistical analysis, making it more difficult to obtain significant differences. We modified the sentence to explain this better: " The yield assessment didn't allow the identification of statistical differences due to the low number of repeats (3-6 plots per treatment; some field margin plots had irregular growth due to uneven water dispersal, weeds, wind, and other reasons)." (lines 549-551).
We also added the following explanation to the Materials and Methods section: "While the treatment and control groups were replicated eight times, some field margin plots had irregular growth (due to uneven water dispersal, weeds, wind, and other reasons). Thus, 3-6 plots per treatment were sampled at the season's end." (lines 347-350).
Line 542-543 For yield, the "outperforming" of the biological treatments (up to 17%) cannot be visually caught in the chart, being not statistically significant. You can say that up to a 17% increase in yield was observed for biological treatments, even if not statistically significant.
The sentence was corrected as suggested, and it now reads: " Most biological treatments had an equal impact as the chemical intervention, increasing the cotton plants' yield by up to 17%, even if not statistically significant (Figure 7A)." (lines 551-553).
Line 543-545 For Symptoms, looking at the statistics, you can only say that the Biological coating+Sp lowered symptoms compared to Chemical coating+D200 and Chemical coating +Sp+D200, but this treatment (Bio+Sp) was not different from all the other treatments, biological or chemical. Words are important.
Reply: You are right. The sentence was edited per your recommendation and now reads: " The biological coating + Sp lowered symptoms compared to chemical coating + D200 and chemical coating + Sp+D200 (up to 31%, p < 0.05, Figure 7B). Still, this treatment (Bio + Sp) was not different from all the other treatments, biological or chemical." (lines 553-556).
546-547 Near harvest, Biocoating+Sp+D400 reduced the pathogen DNA compared to Chemical coating+Sp+D400, but it was not different from all the other treatments (looking at statistics)
Reply: The sentence was edited according to your recommendation: " In the near-harvest sampling, bio-coating + Sp+D400 reduced the pathogen DNA levels by 37% (p < 0.05) compared to chemical coating + Sp+D400 but was statistically equal to all the other treatments (Figure 8)." (lines 556-558).
Line 559 The symptoms evaluation must be described in the Materials and Method section. There it was reported that the disease impact was assessed by counting the number of dead plants. Here in the Figure 7, the symptoms are ranked on a scale 0-20, instead. Please correct or add the necessary information in the MM section.
Figure 7 and 8 The title and legend must be corrected according to the already given recommendations.
Reply: The Y—axis label in Figure 7B (symptoms) was corrected to "Dead plants (no.)". The titles and legends of Figures 7 and 8 were rewritten according to your advice.
Line 550-552 Please explain what the authors intend for traditional chemical intervention. Which data were used to make this t-test? Where are these data? It is not clear.
Reply: The meaning of "traditional chemical intervention" was added to the text. As already explained, the paired t-test analysis was conducted between the bio-coting group of treatments and the chemo-coating group (considering all treatments together in each group). This refers to the field trial data presented in Figure 7B. We modified the text to explain this better:
" While this is being said, a significant difference (p = 0.0433) was found in a paired t-test between the successful bio-chemo integrated management and the traditional chemical intervention (chemical seed coating and applying Azoxystrobin by sprinkling with seeding and/or irrigation during the season), regarding the plants' health. This analysis refers to the field trial symptoms assessment presented in Figure 7B and considers all treatments together in each group." (lines 559-564).
Line 552-554 Where are the percentages of diseased plants? And the number of wilted plants?
Reply: We replaced the phrase "percentage of diseased" with "number of wilting" to describe this better. The numbers of wilted plants are presented in Figure 7B. The sentence now reads: " Also, a significant negative correlation (p = 0.0016) was found between the number of wilting plants (Figure 7B) and the yield (Figure 7A)." (lines 564-565).
Table 8. Move this table among Supplementary materials, correct the Title and note as above recommended
Reply: Table 8 was edited per your suggestion, but we prefer to leave it in the main text (see explanation in Table 6).
Line 587-588 use the acronyms and cite the Figure, to look on the left and on the right
Reply: Corrected as advised.
Line 590 -594 This is not correct. Table 8 reports the mean values calculated over all the plots that received the same treatment, I guess, thus the values of 89% and 69% are mean and not the individual values of the single plots 44 and 53. If the authors want to compare these two plots, they should compare the individual values of disease degree and aerial index of these plots.
Reply: You are right. We rephrase the sentence: "Those two treatments had 89% (bio-coating + Sp) and 69% (bio-coating + D200) symptoms degree in the health estimation (Table 8)." (lines 608-609).
Figure 9 Adjust the title: Aerial imaging of the commercial field trial at 162 days post-sowing, obtained by aerial remote sensing: high-resolution visible range (RGB, A), green–red vegetation index (GRVI, B), and thermal imaging (C). Two adjacent representative plots in the middle of the field are shown: plot 44 on the left treated with Trichoderma spp. mix seed coating + Azoxystrobin sprinkling (Sp), and plot 53 on the right, treated with Trichoderma spp. mix seed coating + Azoxystrobin irrigation at low dose treatment (D200).
Comments to the figure should be mentioned in the text, not in the legend
Reply: Thank you. Corrected as advised.
Table 9 Correct the title and the note of the table as suggested for the other tables. These data are not duplicated by a bar chart, like in the cases above, thus the table can remain here. What about data related to RGB? They are missing in the table.
Reply: The Table 9 was edited according to your suggestion. The RGB image can provide a visual impression, but the data are not quantitative. Yet, the RGB image was processed to provide the GRVI (green-red vegetation index) index based on the green-red ratio. The green wavelengths are more sensitive to high chlorophyll levels since they are less absorbed by chlorophyll a and b, unlike the blue, red, and near-infra-red wavelengths. Therefore, GRVI can detect changes in chlorophyll levels at the leaf scale and is suitable for monitoring plants' development, stresses, and diseases, as conducted here.
The discussion is too long and not very focused on the results obtained. Some paragraphs can be deleted, because are too general. Try to reduce the length. See also below.
Reply: We followed your advice to shorten the Discussion section and make it more focused and more precise.
Line 616 correct condition into disease.
Reply: Corrected as advised.
Line 618-621 Delete.
Reply: Deleted as suggested.
Line 632-642 please summarize.
Reply: Summarized as advised.
Line 653 Taking together, the current etc..
Reply: Corrected as advised.
Line 649-650 correct into: it lessens the impact of chemical treatments on health and environment.
Reply: Corrected as advised.
Line 655-656 It is not supported by the statistic analysis. It was not evident.
Reply: The sentence was rephrased for more accuracy: "Under field conditions, the biological treatments provided crop protection comparable to conventional methods (chemical dressings and Azoxystrobin applications throughout the growing season)." (lines 667-669).
Conclusion. Line 762-763 Correct into: Both trials aimed at evaluating the effectiveness of an integrated biological/chemical approach, i.e. based on biological seed coating with Trichoderma spp. followed by the application of the fungicide Azoxystrobin, in comparison to only biological seed coating, or a completely chemical approach, based on only seed coating or in combination with Azoxystrobin.
Reply: Corrected as advised. Thank you.
Line 764- 767 These statements are not statistically supported, thus I recommend the authors to drive conclusions based on the statistical significance. They must rewrite the sentences, according to the statistical analysis results.
In general, the authors must seriously reflect on the real results of this work and not be driven by what they would have liked to obtain, that is the hypothetical superiority of the integrated approach (biological+chemical). This superiority does not come out from the results of this work, based on one trial in pots and one in field. Maybe they need to repeat the trials.
Reply: We agree and refocus the writing so it will reflect better the outcome of the results: "At this date, biological or chemical seed coating alone or those treatments combined with the sprinkling application (Sp) significantly improved shoot weight compared to treatments combining high doses of chemical preparations." (lines 781-783).
It could be instead interesting to compare the performance of the biological or integrated approach vs the completely chemical.
Reply: Indeed. The following sentence was added to the conclusion instead of the former less precise sentence: "In the field, a significant advantage to the plants' health was found in the bio-chemo integrated management over the chemical intervention (considering all treatments in each category together)." (lines 783-786).
In pots, at 29 days, comparing biological coating vs chemical coating, either alone or within each treatment (i.e., the green vs the yellow bars) I can see that the green bars were always NOT different from the yellow (i.e. biocoating not different from chemical coating and bio-chemical integration not different from chemical) for all the growth parameters. Unique exceptions: shoot height and number of leaves are stimulated by biocoating+Sp+D200 compared to the corresponding chemical coating+Sp+D200.
At 52 days, it is the same, but the unique exception is valid only for shoot height.
For root infection, at 29 days, no differences. At 52 days, chemical coating+D400 is more protective than the corresponding biocoating+D400. The same occurs for Chemical+Sp+400 which is superior to the corresponding biocoating+Sp+D400.
In the field, no differences at all among the treatments for yield, and no differences among bio+chem and chem for symptoms at 151 days. This is interesting because it means that the seed coating only with Trichoderma provides enough protection to sustain yield and control CRD, as well as all the other treatments differently combined! A great result from an economic and environmental point of view. About the root infection, no differences between biocoating and chem coating or bio+chem and completely chem approach at 74 days. At 151 days it is the same, except for biocoat+Sp+400 which was superior to the corresponding chem+Sp+D400
Thus, I will finally suggest shifting the work in this direction, instead of trying to find the best option among the treatments, because the results do not allow to find the best option.
Reply: We agree with this point of viewing the result. We followed this logic and rewrote the main findings in the conclusion section accordingly:
"In the growing room trial, comparing the bio-based coating treatments versus their parallel chemical coating partners in most cases resulted in similar outcomes on both sampling dates. Unique exceptions are the shoot height and number of leaves. These are stimulated by bio-coating in combination with Azoxystrobin sprinkling in the sowing strip (Sp) and low dosage irrigation (D200) compared to the corresponding chemical coating+Sp+D200. At 52 days, this result was the same, but the unique exception is valid only for shoot height. At this date, biological or chemical seed coating alone or those treatments combined with the sprinkling application (Sp) significantly improved shoot weight compared to treatments combining high doses of chemical preparations. In the field, a significant advantage to the plants' health was found in the bio-chemo integrated management over the chemical intervention (considering all treatments in each category together). Regarding yield, no differences were measured among the bio- and chemo-based treatments. This means that the seed coating only with Trichoderma provides enough protection to sustain yield and control CRD, as well as all the other treatments differently combined." (lines 775-789).
Round 3
Reviewer 3 Report
The manuscript has been quite improved, being most of my suggestions included in this new version.
The authors were asked to rewrite some parts of the Results, taking into account the statistics. I appreciate the new version but I regret to say that there are some more revisions to make before the manuscript can be published.
A general note about Tables 6, 7 and 8.
One of the most basic rules of academic writing is that the same data cannot be presented as both a table and a figure. This would mean duplication of content. An author should choose the best form of representing his data, in figure or table. Normally a figure is more immediate. However, if an author feels that the numerical data are also important, he can include a table as supplementary material. In this case, the author must add a footnote after the figure mentioning that a detailed table with all the data has been included as supplementary material.
Please also note that tables 6,7 and 8 do not include statistics, and thus they are misleading. The authors themselves were misled when commenting on their results by looking at the tables instead of the statistics reported in the Figures.
So, I strongly suggest moving these tables among the Supplementary materials. I have revised this manuscript version, according to this situation.
Another important issue is to revise the abstract according to the results presented in this final version of the manuscript.
Here there are my further suggestions, hoping that they will be useful to the authors to refine definitely the manuscript
Line 200-209 Be more synthetic in the description of the treatments listed in Table 2, avoiding repetitions and discussion, and following the list order. Delete from line 200 up to 209, from “Since the standard”…..to “crop production” and substitute the paragraph with:
“The chemical seed coating was performed by Israel Seeds Company with a standard fungicide mix (thiram, captan, and metalaxyl-M), which does not protect from CRD. The biological seed coating was performed with a mix of spores, mycelium fragments, and extrolites from the different Trichoderma cultures, as described in par. 2.2.2. The treatments were compared to two additional mock groups of non-infected plants that underwent solely biological or chemical seed coating.
Line 209 correct 96 into 98 pots (because 14 groups x 7 replicates =98)
Line 216-212 Footnote: Avoid materials and methods details and do not refer to another table. Thus correct the footnote into:
1standard mix of thiram, captan, carboxin, and Metalaxyl-M; 2mix of fungal spores, mycelium fragments, and extrolites from Trichoderma T7107, T7407, P1 cultures; 3AS: Azoxystrobin; 4DDW: double-distilled water
Figure 1 As I pointed out in my previous report, this figure does not add information to the trial architecture already fully described in Table 2. Anyway, since this figure was inserted to meet the request of another referee, I would suggest moving it to Supplementary materials. The Figure clarity has been improved, however, the caption is verbose and contains materials and methods details, besides referring to Table 2, thus it should be shortened and corrected like this:
Figure S1. Experimental design of the pot assay showing the different treatments applied after sowing in substrate infected with Macrophomina phaseolina. Two types of seed coating were compared, biological (Trichoderma mix) or chemical (a mixture of thiram, captan, carboxin, and metalaxyl-M), solely or combined with the following five different applications of the fungicide Azoxystrobin (AS): by sprinkling at sowing, by irrigation (two dosages) on 10- and 21-day post sowing, or by sprinkling + irrigation. Healthy, non-infected control groups (which also had biological or chemical seed coating) were included. Every experimental group included seven biological replications, for a total of 98 pots. (Do not cite the field trial)
Besides, the character size of the AS dosage is too small, thus quite unreadable and it should be enlarged
Moreover, it is necessary to cite the Figure together with the citation of Table 2, at line 199 like this: Such treatments are listed in Table 2 and shown in Figure S1.
Line 236 Correct into The Trichoderma isolates
Line 248 Delete “The dry seeds were taken to the field to be sown”
Line 251 correct into “had adhered”
Line 267 correct developed into grown
Line 316-322 It is not fine that the description of the treatments arrives at the end of the paragraph. It is opportune to move the entire block, from “As in the growth room….from sowing.” ahead, at line 307, before the sentence “Each treatment group included eight replications etc..”
Line 428-433 The comparison is done over the healthy control, while it should have done over the infected control. Anyway, I already recommended focusing on statistically significant results only. The 21% increase in the root weight compared to the healthy control is not statistically significant. This means that it could be only an apparent increase. The authors continue to comment data from the Table ignoring the statistics presented in the Figure. The paragraph must be corrected into:
“Looking at the growth parameters measured on day 29 post-sowing (Figure 2A-C), the integrated approach never performed worse than the chemical approach, according to statistical analysis (Figure 3A-F).
Line 433-435 “In contrast, negative effects are highlighted on shoot weight and height, number of leaves due to Sp+D200 and sp+D400 treatments combined with chemical seed coating (p < 0.05)”. Very good observation! This is indeed supported by statistics. Please add (Figure 3A,B and E)
Line 437 add “chemical” between “above” and “treatments”
Line 437-439 delete the entire sentence, the meaning is not clear. In non-infected plants, biological coating gave lower shoot weight only, compared to chemical coating. The other growth parameters were not different, based on the coating type.
Line 439-441 The authors write: “A significant improvement over the bio-coating healthy control can be seen in the Trichoderma species' mixture seed coating alone (SC only) shoot weight (44%, p < 0.005, Figure 3A)”.
This means that plants in infected soil are higher than plants in healthy soil, even though both are treated with Trichoderma. How can you explain this? Did something go wrong with the healthy controls? I would not highlight this phenomenon. It is also not relevant for the disease control. Delete this sentence.
Line 443-444 The authors write: The biological SC also achieved statistically discernible improvement in the root weight… This is false. For root weight, biological SC has “ab”, thus it is not different from all the other treatments and controls. Thus delete this statement and adjust the sentence accordingly
Line 447-450 The authors write: Evidentially, all chemical treatments that included both Azoxystrobin sprinkling and irrigation (Sp + D200 and Sp + D400) led to 15-29% growth suppression (Table 7), with significant reduction (p < 0.05, Figure 3 A- B, E) in the shoot development when combined with chemo seed coating most likely due to a phytotoxicity resulting from overdose”.
1)The authors did not say compared to what treatment. 2) they did not say for which parameters. 3)Please, do not cite percentages that cannot be found in the figures or tables. 4) The citation of Table 7 is wrong because it refers to DNA.
Compared with the chemical SC only, the significant reduction in shoot weight, shoot height and number of leaves was supported by the statistics only for Sp+200. Compared with the healthy control, no significant reductions are highlighted. Thus correct line 447-450 into: The chemical coating combined with Azoxystrobin sprinkling and irrigation at low dosage (Sp + D200) significantly reduced the shoot development (shoot weight, shoot height, and leaves number) compared to SC only (Figure 3A, B, E)
Delete “most likely due to a phytotoxicity resulting from overdose(p < 0.05, Figure 3 A- B, E)” because it is a discussion and because the highest dose D400 did not result different from Sc only, thus the hypothesis of phytoxicity is not supported.
Line 450-452 The authors write: “Interestingly, in the biological seed-coating plants, the double Azoxystrobin treatment (Sp + D200) was significantly less harmful (p << 0.05)”. For which parameter? Differences are significant only for the number of leaves, thus correct into : Interestingly, for the number of leaves, the treatment (Sp + D200) was significantly less harmful when combined with the biological seed coating than with the chemical coating (p < 0.05).(Figure 3E)
Figure 3 caption: Correct the title into Growth parameters and survival rate of the plants after the different treatments in the pot assay on day 29. The same captions is suggested for Figure 4, substituting obviously on day 29 with on day 52
Line 455-456 correct “phonological stage (number of leaves, E), and the serving plants' percentage (D)” into number of leaves (E), and survival rate (F)” (the same corrections are suggested for Figure 4)
Line 466 Correct into: At the end of the experiment, on day 52 (figure 2E, F)
Line 468-473 All these statements are not statistically supported. There are no significant increases, no improvements, no impacts. Stay focused on the statistics, as you stated that the results were similar to day 29, with less statistical significance. Do not base the comment on data reported in the table. Delete all these sentences from line 468-471.
Line 473-477. The comparison term is missing and Sp+400 did not show any significant effect of reduction. Please rephrase the sentence like this: As for day 29, the treatments Sp + D200, combining chemical coating with Azoxystrobin sprinkling and irrigation, exerted a negative effect on plant development, significantly reducing shoot weight and height, and leaves number compared to chemical SC only (Figure 4 A,B and E).
Line 497 add “are” before “reported”
Line 498 Regard day 52 data
Line 502-504.The author say: “In agreement with the growth results, combining chemical seed coating and Azoxystrobin sprinkling (Sp) led to the greatest pathogen repression on day 29 (Table 7)” Again the authors ignore the statistics reported in Figure 5 and comment data from table. This treatment (Sp) was marked bcd and did not differ from all the other treatments (except for Chemical SC+Sp+D400, which was marked with “a”) and must not be pointed out as the one that led to the greatest repression. Thus, delete the sentence.
Line 504-507 The authors say: “Unexpectedly, this treatment was less effective on day 52, whereas the bio-based integrated treatment (bio-coating + Azoxystrobin sprinkling, D200) excelled. This result fits the roots' growth promotion outcome of this management, which excelled on that date”.
Delete all this. It is not true that the bio-coating+ D200 “excelled”. In Figure 5 it has been marked with “cd” thus it performed not differently from all the other treatments that were marked with “c” or “d”. I had already recommended the authors not to base their statement on the absolute values reported in the Tables.
I would instead comment that, for the root infection, as already observed for the plant growth parameters, the biological and integrated approach performed as well as the chemical approach on day 29, while on day 52 the chemical SC+D400 was more protective than the corresponding biocoating+D400. The same occurred for chemical SC +Sp+400 which was superior to the corresponding biocoating+Sp+D400.
Line 551 Delete “Most”. The biological treatments etc.
Line 558 Figure 8B
Line 560 delete “successful”
Line 568-570 This “summarizing” is not supported by the results and must be deleted. Treatments did not influence the yield at all; bio-coating +Sp, showed some symptom reduction if compared only to chem coating+D200 or chem coating+Sp+D200, but it was not different from all the other treatments; bio-coat+Sp+D400 reduced pathogen DNA in the root if compared to the corresponding chem+Sp+400, but it did not differ from all the other treatment. Thus, summarizing is quite impossible.
I would suggest a sentence like: Generally, the integrated approach in the field performed not differently than the chemical approach, considering yield, plant symptoms, and root infection. Some integrated treatments appeared promising in reducing CRD symptoms, like biochemical coating combined with Azoxystrobin sprinkling at sowing (Figure 7B)
Figure 7A: Important: please check the measurement unit on the Y axis. I think it should be kg/m2
Line 771 change “under a growth room and commercial scale”” into “in a growth room and at commercial scale”
Line 773 add “was” before “tested”
Line 775 in the growth room trial
Line 777-783 The author reported here what I observed in my previous report, but the authors did not include them in the new version of the manuscript, so they cannot report them in the Conclusion. On this point, in the Results section they made different observations and these should be reported here (see lines Line 433-435 “In contrast, negative effects are highlighted on shoot weight and height, number of leaves due to Sp+D200 and sp+D400 treatments combined with chemical seed coating (p < 0.05)”.Thus I suggest substitute lines 777-783 utilizing their own sentences: “In contrast, negative effects were highlighted on the plant development, when applying the hardest chemical approach based on seed coating followed by repeated fungicide application”. I would not specify day 29 or 52, since these are conclusions.
Line 783 Insert a sentence that recall the similarity of the results obtained in growth room and field, like this: “Generally, the integrated approach in the field performed not differently than the chemical approach, considering yield, plant symptoms, and root infection. Some integrated treatments appeared promising in reducing CRD symptoms, like biochemical coating combined with Azoxystrobin sprinkling at sowing.
Line 787-789 I would be cautious since the field experiment was not repeated. Use “This could even mean that …but should be verified repeating the field trial.
Finally and importantly, the authors must correct the abstract based on the statistically significant results reported in the final manuscript version.

Author Response
Responses to Reviewer 3's comments
We are grateful to the reviewer for investing substantial effort, which undoubtedly contributed to this manuscript. Thank you for the detailed and wise advice and the guidance you provided in improving the article and preparing it for publication. Your contribution is greatly appreciated.
Major comments
The manuscript has been quite improved, being most of my suggestions included in this new version.
The authors were asked to rewrite some parts of the Results, taking into account the statistics. I appreciate the new version but I regret to say that there are some more revisions to make before the manuscript can be published.
A general note about Tables 6, 7 and 8.
One of the most basic rules of academic writing is that the same data cannot be presented as both a table and a figure. This would mean duplication of content. An author should choose the best form of representing his data, in figure or table. Normally a figure is more immediate. However, if an author feels that the numerical data are also important, he can include a table as supplementary material. In this case, the author must add a footnote after the figure mentioning that a detailed table with all the data has been included as supplementary material.
Please also note that tables 6,7 and 8 do not include statistics, and thus they are misleading. The authors themselves were misled when commenting on their results by looking at the tables instead of the statistics reported in the Figures.
So, I strongly suggest moving these tables among the Supplementary materials. I have revised this manuscript version, according to this situation.
Reply: Thank you for helping us in improving this manuscript. Tables 6, 7, and 8 moved from the main body text to the Supplementary materials. We also accept all your other suggestions and corrections, as detailed below.
Another important issue is to revise the abstract according to the results presented in this final version of the manuscript.
Reply: we revised the abstract as suggested. The abstract was corrected to match the analysis of the results and the concluding section. It is now focused on those results that gain statistical significance.
Here there are my further suggestions, hoping that they will be useful to the authors to refine definitely the manuscript.
Reply: thank you. All your suggestions were addressed thoroughly, as detailed below, and the changes are reflected in the revised manuscript version.
Specific Comments:
Line 200-209 Be more synthetic in the description of the treatments listed in Table 2, avoiding repetitions and discussion, and following the list order. Delete from line 200 up to 209, from "Since the standard"…..to "crop production" and substitute the paragraph with:
"The chemical seed coating was performed by Israel Seeds Company with a standard fungicide mix (thiram, captan, and metalaxyl-M), which does not protect from CRD. The biological seed coating was performed with a mix of spores, mycelium fragments, and extrolites from the different Trichoderma cultures, as described in par. 2.2.2. The treatments were compared to two additional mock groups of non-infected plants that underwent solely biological or chemical seed coating.
Reply: Corrected as advised.
Line 209 correct 96 into 98 pots (because 14 groups x 7 replicates =98).
Reply: Corrected as advised.
Line 216-212 Footnote: Avoid materials and methods details and do not refer to another table. Thus correct the footnote into:
1standard mix of thiram, captan, carboxin, and Metalaxyl-M; 2mix of fungal spores, mycelium fragments, and extrolites from Trichoderma T7107, T7407, P1 cultures; 3AS: Azoxystrobin; 4DDW: double-distilled water.
Reply: Corrected as advised.
Figure 1 As I pointed out in my previous report, this figure does not add information to the trial architecture already fully described in Table 2. Anyway, since this figure was inserted to meet the request of another referee, I would suggest moving it to Supplementary materials. The Figure clarity has been improved, however, the caption is verbose and contains materials and methods details, besides referring to Table 2, thus it should be shortened and corrected like this:
Figure S1. Experimental design of the pot assay showing the different treatments applied after sowing in substrate infected with Macrophomina phaseolina. Two types of seed coating were compared, biological (Trichoderma mix) or chemical (a mixture of thiram, captan, carboxin, and metalaxyl-M), solely or combined with the following five different applications of the fungicide Azoxystrobin (AS): by sprinkling at sowing, by irrigation (two dosages) on 10- and 21-day post sowing, or by sprinkling + irrigation. Healthy, non-infected control groups (which also had biological or chemical seed coating) were included. Every experimental group included seven biological replications, for a total of 98 pots. (Do not cite the field trial).
Besides, the character size of the AS dosage is too small, thus quite unreadable and it should be enlarged.
Moreover, it is necessary to cite the Figure together with the citation of Table 2, at line 199 like this: Such treatments are listed in Table 2 and shown in Figure S1.
Reply: The character size of the AS dosage was enlarged. The figure legend was replaced with the suggested text. The Figure was moved to the Supplementary materials. Figure S1 was cited together with the citation of Table 2, as advised.
Line 236 Correct into The Trichoderma isolates.
Reply: Corrected as advised.
Line 248 Delete "The dry seeds were taken to the field to be sown".
Reply: Deleted as advised.
Line 251 correct into "had adhered".
Reply: Corrected as advised.
Line 267 correct developed into grown.
Reply: Corrected as advised.
Line 316-322 It is not fine that the description of the treatments arrives at the end of the paragraph. It is opportune to move the entire block, from "As in the growth room….from sowing." ahead, at line 307, before the sentence "Each treatment group included eight replications etc.."
Reply: The paragraph location was corrected as advised.
Line 428-433 The comparison is done over the healthy control, while it should have done over the infected control. Anyway, I already recommended focusing on statistically significant results only. The 21% increase in the root weight compared to the healthy control is not statistically significant. This means that it could be only an apparent increase. The authors continue to comment data from the Table ignoring the statistics presented in the Figure. The paragraph must be corrected into:
"Looking at the growth parameters measured on day 29 post-sowing (Figure 2A-C), the integrated approach never performed worse than the chemical approach, according to statistical analysis (Figure 3A-F).
Reply: Corrected as advised.
Line 433-435 "In contrast, negative effects are highlighted on shoot weight and height, number of leaves due to Sp+D200 and sp+D400 treatments combined with chemical seed coating (p < 0.05)". Very good observation! This is indeed supported by statistics. Please add (Figure 3A,B and E).
Reply: Thank you. We added (Figure 2A, B, and E).
Line 437 add "chemical" between "above" and "treatments".
Reply: Added as recommended.
Line 437-439 delete the entire sentence, the meaning is not clear. In non-infected plants, biological coating gave lower shoot weight only, compared to chemical coating. The other growth parameters were not different, based on the coating type.
Reply: Deleted as advised.
Line 439-441 The authors write: "A significant improvement over the bio-coating healthy control can be seen in the Trichoderma species' mixture seed coating alone (SC only) shoot weight (44%, p < 0.005, Figure 3A)".
This means that plants in infected soil are higher than plants in healthy soil, even though both are treated with Trichoderma. How can you explain this? Did something go wrong with the healthy controls? I would not highlight this phenomenon. It is also not relevant for the disease control. Delete this sentence.
Reply: Trichoderma species can act as biostimulators and enhance the plant's growth by interacting with the plant's micorhiza or by other mechanisms. See: López-Bucio, José, Ramón Pelagio-Flores, and Alfredo Herrera-Estrella. "Trichoderma as biostimulant: exploiting the multilevel properties of a plant beneficial fungus." Scientia horticulturae 196 (2015): 109-123.
We accepted your suggestion and deleted this sentence.
Line 443-444 The authors write: The biological SC also achieved statistically discernible improvement in the root weight… This is false. For root weight, biological SC has "ab", thus it is not different from all the other treatments and controls. Thus delete this statement and adjust the sentence accordingly.
Reply: Right. The reference to the root weight was deleted, and the sentence was adjusted.
Line 447-450 The authors write: Evidentially, all chemical treatments that included both Azoxystrobin sprinkling and irrigation (Sp + D200 and Sp + D400) led to 15-29% growth suppression (Table 7), with significant reduction (p < 0.05,
Figure 3 A- B, E) in the shoot development when combined with chemo seed coating most likely due to a phytotoxicity resulting from overdose".
1)The authors did not say compared to what treatment. 2) they did not say for which parameters. 3)Please, do not cite percentages that cannot be found in the figures or tables. 4) The citation of Table 7 is wrong because it refers to DNA.
Compared with the chemical SC only, the significant reduction in shoot weight, shoot height and number of leaves was supported by the statistics only for Sp+200. Compared with the healthy control, no significant reductions are highlighted. Thus correct line 447-450 into: The chemical coating combined with Azoxystrobin sprinkling and irrigation at low dosage (Sp + D200) significantly reduced the shoot development (shoot weight, shoot height, and leaves number) compared to SC only (Figure 3A, B, E)
Delete "most likely due to a phytotoxicity resulting from overdose(p < 0.05, Figure 3 A- B, E)" because it is a discussion and because the highest dose D400 did not result different from Sc only, thus the hypothesis of phytoxicity is not supported.
Reply: We accepted your remarks and corrected the paragraph according to your advice.
Line 450-452 The authors write: "Interestingly, in the biological seed-coating plants, the double Azoxystrobin treatment (Sp + D200) was significantly less harmful (p << 0.05)". For which parameter? Differences are significant only for the number of leaves, thus correct into : Interestingly, for the number of leaves, the treatment (Sp + D200) was significantly less harmful when combined with the biological seed coating than with the chemical coating (p < 0.05).(Figure 3E).
Reply: Corrected as advised.
Figure 3 caption: Correct the title into Growth parameters and survival rate of the plants after the different treatments in the pot assay on day 29. The same captions is suggested for Figure 4, substituting obviously on day 29 with on day 52.
Reply: Corrected as advised.
Line 455-456 correct "phonological stage (number of leaves, E), and the serving plants' percentage (D)" into number of leaves (E), and survival rate (F)" (the same corrections are suggested for Figure 4).
Reply: Corrected as advised.
Line 466 Correct into: At the end of the experiment, on day 52 (figure 2E, F).
Reply: Corrected as advised.
Line 468-473 All these statements are not statistically supported. There are no significant increases, no improvements, no impacts. Stay focused on the statistics, as you stated that the results were similar to day 29, with less statistical significance. Do not base the comment on data reported in the table. Delete all these sentences from line 468-471.
Reply: Agreed. Deleted as advised.
Line 473-477. The comparison term is missing and Sp+400 did not show any significant effect of reduction. Please rephrase the sentence like this: As for day 29, the treatments Sp + D200, combining chemical coating with Azoxystrobin sprinkling and irrigation, exerted a negative effect on plant development, significantly reducing shoot weight and height, and leaves number compared to chemical SC only (Figure 4 A,B and E).
Reply: Corrected as advised.
Line 497 add "are" before "reported".
Reply: Added as advised.
Line 498 Regard day 52 data.
Reply: Corrected as advised.
Line 502-504.The author say: "In agreement with the growth results, combining chemical seed coating and Azoxystrobin sprinkling (Sp) led to the greatest pathogen repression on day 29 (Table 7)" Again the authors ignore the statistics reported in Figure 5 and comment data from table. This treatment (Sp) was marked bcd and did not differ from all the other treatments (except for Chemical SC+Sp+D400, which was marked with "a") and must not be pointed out as the one that led to the greatest repression. Thus, delete the sentence.
Reply: Agreed. Deleted as advised.
Line 504-507 The authors say: "Unexpectedly, this treatment was less effective on day 52, whereas the bio-based integrated treatment (bio-coating + Azoxystrobin sprinkling, D200) excelled. This result fits the roots' growth promotion outcome of this management, which excelled on that date".
Delete all this. It is not true that the bio-coating+ D200 "excelled". In Figure 5 it has been marked with "cd" thus it performed not differently from all the other treatments that were marked with "c" or "d". I had already recommended the authors not to base their statement on the absolute values reported in the Tables.
I would instead comment that, for the root infection, as already observed for the plant growth parameters, the biological and integrated approach performed as well as the chemical approach on day 29, while on day 52 the chemical SC+D400 was more protective than the corresponding biocoating+D400. The same occurred for chemical SC +Sp+400 which was superior to the corresponding biocoating+Sp+D400.
Reply: The paragraph was corrected as advised.
Line 551 Delete "Most". The biological treatments etc.
Reply: Deleted as advised.
Line 558 Figure 8B.
Reply: Corrected as advised.
Line 560 delete "successful".
Reply: Deleted as advised.
Line 568-570 This "summarizing" is not supported by the results and must be deleted. Treatments did not influence the yield at all; bio-coating +Sp, showed some symptom reduction if compared only to chem coating+D200 or chem coating+Sp+D200, but it was not different from all the other treatments; bio-coat+Sp+D400 reduced pathogen DNA in the root if compared to the corresponding chem+Sp+400, but it did not differ from all the other treatment. Thus, summarizing is quite impossible.
I would suggest a sentence like: Generally, the integrated approach in the field performed not differently than the chemical approach, considering yield, plant symptoms, and root infection. Some integrated treatments appeared promising in reducing CRD symptoms, like biochemical coating combined with Azoxystrobin sprinkling at sowing (Figure 7B).
Reply: The paragraph was corrected as advised.
Figure 7A: Important: please check the measurement unit on the Y axis. I think it should be kg/m2.
Reply: You are right. We apologize for this typo. Corrected to kg/m2.
Line 771 change "under a growth room and commercial scale"" into "in a growth room and at commercial scale".
Reply: Corrected as advised.
Line 773 add "was" before "tested".
Reply: Corrected as advised.
Line 775 in the growth room trial.
Reply: Corrected as advised.
Line 777-783 The author reported here what I observed in my previous report, but the authors did not include them in the new version of the manuscript, so they cannot report them in the Conclusion. On this point, in the Results section they made different observations and these should be reported here (see lines Line 433-435 "In contrast, negative effects are highlighted on shoot weight and height, number of leaves due to Sp+D200 and sp+D400 treatments combined with chemical seed coating (p < 0.05)".Thus I suggest substitute lines 777-783 utilizing their own sentences: "In contrast, negative effects were highlighted on the plant development, when applying the hardest chemical approach based on seed coating followed by repeated fungicide application". I would not specify day 29 or 52, since these are conclusions.
Reply: We agree and have corrected the paragraph per your recommendations.
Line 783 Insert a sentence that recall the similarity of the results obtained in growth room and field, like this: "Generally, the integrated approach in the field performed not differently than the chemical approach, considering yield, plant symptoms, and root infection. Some integrated treatments appeared promising in reducing CRD symptoms, like biochemical coating combined with Azoxystrobin sprinkling at sowing.
Reply: The suggested paragraph was added to the text. Thank you.
Line 787-789 I would be cautious since the field experiment was not repeated. Use "This could even mean that …but should be verified repeating the field trial.
Reply: Corrected as advised.
Finally and importantly, the authors must correct the abstract based on the statistically significant results reported in the final manuscript version.
Reply: The abstract was corrected to match the analysis of the results and the concluding section. It is now focused on those results that gain statistical significance. Thank you for helping us improve this manuscript.